# Data-driven segmentation of cortical calcium dynamics

**Sydney C. Weiser** [1☯], **Brian R. Mullen** [1☯]*, **Desiderio Ascencio** [2¤], **James B. Ackman** [1]*

1 Department of Molecular, Cell, and Developmental Biology, University of California Santa Cruz, Santa Cruz, California, United States of America, 2 Department of Psychology, University of California Santa Cruz, Santa Cruz, California, United States of America

☯ These authors contributed equally to this work.
¤ Current address: Biology and Biological Engineering, California Institute of Technology, Pasadena, California, United States of America
* brmullen@ucsc.edu (BRM); james.ackman@gmail.com (JBA)

**Data Availability Statement:** The core pipeline with proper documentation showcasing its capabilities will be found at https://pypi.org/project/seas/ The machine learning code and metric

## Abstract

Demixing signals in transcranial videos of neuronal calcium flux across the cerebral hemispheres is a key step before mapping features of cortical organization. Here we demonstrate that independent component analysis can optimally recover neural signal content in widefield recordings of neuronal cortical calcium dynamics captured at a minimum sampling rate of $1.5 \times 10^6$ pixels per one-hundred millisecond frame for seventeen minutes with a magnification ratio of 1:1. We show that a set of spatial and temporal metrics obtained from the components can be used to build a random forest classifier, which separates neural activity and artifact components automatically at human performance. Using this data, we establish functional segmentation of the mouse cortex to provide a map of ~115 domains per hemisphere, in which extracted time courses maximally represent the underlying signal in each recording. Domain maps revealed substantial regional motifs, with higher order cortical regions presenting large, eccentric domains compared with smaller, more circular ones in primary sensory areas. This workflow of data-driven video decomposition and machine classification of signal sources can greatly enhance high quality mapping of complex cerebral dynamics.

## Author summary

Researchers have been able to record from a large population of neurons across the cortex using calcium indicators in awake, behaving mice; however, many confounding neuronal signals (ie. neurons from different depths) or tissue dynamics (ie. blood flow) influence these recordings. Our custom pipeline utilizes algorithms to identify distinct signals and spatially segment the signal sources into components that can then be characterized. From these components, we show that neuronal signals are distinct from non-neuronal artifacts; further, we are able to remove the artifacts to clean the neuronal signal. The remaining components are spatial segments of the neuronal signals; we use them to create a data-driven map of functional units, which we call domains. We characterize these

generation are available Ackman lab GitHub page (https://github.com/ackmanlab/) All relevant data is uploaded to Dryad https://doi.org/10.7291/D1N96W.

**Funding:** This work was supported by Startup funds from University of California, Santa Cruz, Division of Physical and Biological Sciences, grants from the National Institutes of Health, USA (NIH T32 GM 133391) to S.C.W. and (NIH T32 GM 864620) B.R.M, and by a Hellman Fellows Fund Award to J.B.A. Funding for D.A. was provided by the UCSC Maximizing Access to Research Careers (MARC) program (T32-GM007910) and the UCSC Initiative for Maximizing Student Development (IMSD) (R25-GM058903). The funders had no role in study design, data collection and analysis, decision to publish, or preparation of the manuscript.

**Competing interests:** The authors have declared that no competing interests exist.

domains between subsequent recordings and show that they are highly similar within the same animal, given a long enough recording. This data-driven map can provide information about the limitations one can make from the specific recordings. Furthermore, it will enhance our ability to understand the effects on functional maps from animals that do not have a reference map, including mapping functional changes in development, differences between genetically mutated or varying strains of mice.

This is a *PLOS Computational Biology* Methods paper.

## Introduction

Optical techniques have long been used to monitor the functional dynamics in sets of neuronal elements ranging from isolated invertebrate nerve fibers [1,2] to entire regions of mammalian visual cortex in vivo [3–5]. Imaging calcium flux with calcium sensors [6,7] allows for monitoring transcranial neural activity across the cortical surface of the mouse with high enough spatiotemporal resolution to identify sub-areal networks of the neocortex [8,9]. These techniques have the potential to map supracellular group function at unprecedented resolution and scale across the neocortical sheet in awake behaving mice; however, identifying neural signals from calcium imaging sessions is challenging due to numerous confounding signal sources.

Wide-field cortical calcium imaging provides a unique combination of spatially and temporally resolved dynamics across the cortical surface, with scale ranging from complex activation patterns in high-order circuits to discrete activations hundreds of micrometers in diameter to whole cortical lobe activity patterns [8,9]. However, these techniques are affected by issues common to all optical imaging recording. Body or facial movements can create large fluctuations in autofluorescence of the brain and blood vessels, which produce significant artifacts in the data. Vascular artifacts are commonly seen due to vasodynamics and the resulting changes in blood flow required to meet the energy demands of surrounding tissue. Fluid exchange between vascular and neural tissue causes cortical hemodynamics, resulting in region specific changes of optical properties among cerebral lobes [10]. Further, though the skull is fixed to a specific location during the experiment, slight brain movements occur within the cranium, thereby influencing the recordings. Any optical property differences that originate from the experimental preparation may be highlighted in the dataset as signal due to changes in tissue contrast.

Recording parameters need to be taken into consideration to ensure high signal quality and understanding the limitations of wide-field calcium dynamics need to be addressed quantitatively. Researchers have recorded wide field calcium dynamics at frame rates ranging from 5-100Hz [9,11,12]. In addition, spatial resolution varies between different researchers' setups, but is typically in the range of 256x256 to 512x512 pixels (0.06 to 0.2 megapixels) for the entire cortical surface, and is often further spatially reduced for processing [9,11,13]. Election of resolution is often dependent on the video observer's perceived quality of the data or available computational resources, rather than a quantified comparison of signal content. Further, considerations need to be taken on the signal source to understand the impact on signal quality [14].

It is common to use sensory stimulation to identify specific regions in the neocortex, then align a reference map based on the location of these defined regions [13,15,16]. Even if these

maps are reliable for locating primary sensory areas, they often lack specificity for higher order areas, or even completely lack sub-regional divisions. This is especially true in areas with a high degree of interconnectedness and overlapping functionality, such as motor cortex [17]. Moreover, there is evidence that the shape and location of higher order regions can vary from subject to subject [18,19]. Improper map alignment or misinformed regional boundaries can lead to a loss in dynamic range between signals across a regional border. In order to extract the most information from a recorded dataset, the level of parcellation must reflect the quality and sources present within the data. Collectively, these considerations demonstrate that a flexible data-driven method is necessary; furthermore, it must also respect functional boundaries of the cortex and be sensitive to age, genotype and individual variation.

Eigendecompositions can be used to identify and filter components of signal [20–22], present a flexible method of filtering that is not hardware dependent, and be applied to any video dataset regardless of the recording hardware or parameters. An eigendecomposition pipeline that utilizes non-negative matrix factorization has been developed to explore the functional activities across wide-field imaging of the cortex, but this method is limited by the use of a reference map and cannot separate artifact signals from neural activation [23]. Independent Component Analysis (ICA) [24] has the potential to overcome these limitations and has been previously applied to fMRI and EEG data with varying success; for example, identifying both intrinsic connectivity networks rather than individual areas and artifacts that represent large-scale effects rather than spatially localized effects [25–28]. We hypothesize that this is due to the lower density of spatial sampling in fMRI and EEG data.

ICA is a blind source separation algorithm utilized on multichannel data to reveal underlying signal sources. In utilizing this algorithm, we first make the assumption that confounding artifacts and neural calcium signals behave with differing temporal and spatial properties; therefore, artifacts can be removed to result in data that is exclusively neural. ICA has had great result de-mixing resting state fMRI data that requires no a-priori information about the signals of interest [29–31]. Secondly, we assume that the calcium signals collected from structured populations of neurons will produce repetitive and consistent network activation, allowing the algorithm to isolate functional activity based on underlying cytoarchitecture. As such, we can apply ICA decomposition to gain understanding of the functional networks across the cortex.

Here we present an ICA-based workflow that isolates and filters artifacts from calcium imaging videos, with principled exploration of each component to identify each signal source necessary to reduce the contamination resulting from these physiological dynamics. ICA is a nonparametric unsupervised machine learning (ML) technique that can identify each signal source in densely sampled (5.5 million pixels per frame) calcium imaging videos based on their spatially co-activating pixels and temporal properties. The global mean time course was initially subtracted and stored, thereby allowing ICA to decompose each signal distinct from global effects. The decomposition results in hundreds of neural source components per hemisphere that are distinctly de-mixed from artifact source signals. Our concurrent analysis of control wide-field imaging data corroborates the identification of artifact signal sources and gives insight into the structure of neuronal calcium dynamics across neocortex.

We also explore the resolution-dependent effect of signal extraction on ICA quality, and find a quantified increase in ICA signal separation for collecting wide-field calcium imaging at mesoscale resolution. Additionally, using neural components, we generate data-driven maps that are specific to functional borders from individual animals. We use these maps to extract time series from functional regions of the cortex, and show that this method for time series extraction produces a reduced set of time series while optimally representing the underlying signal and variation from the original dataset. Together, these methods provide a set of

optimized techniques for enhanced filtering, segmentation, and time series extraction for
wide-field calcium imaging videos.

## Results

To record neural activity patterns in the cortex of awake behaving adult mice, we transcranially
recorded fluorescence from a mouse that has the genetically encoded calcium indicator,
GCaMP6s, expressed in all neurons under the control of the Snap25 promoter [32]. We expose
and illuminate the cranium with blue wavelength light and capture emitted green light with a
sCMOS camera at high spatial resolution (2160x2560 pixels, 5.5 megapixels; ∼ 6.9 μm/pixel).
To observe the spatiotemporal properties of the recorded activity patterns, we crop the video
to only neural tissue, and compare the change in fluorescence over the mean fluorescence: ΔF/
F over time (Fig 1A and 1B). In order to identify components associated with artifacts and

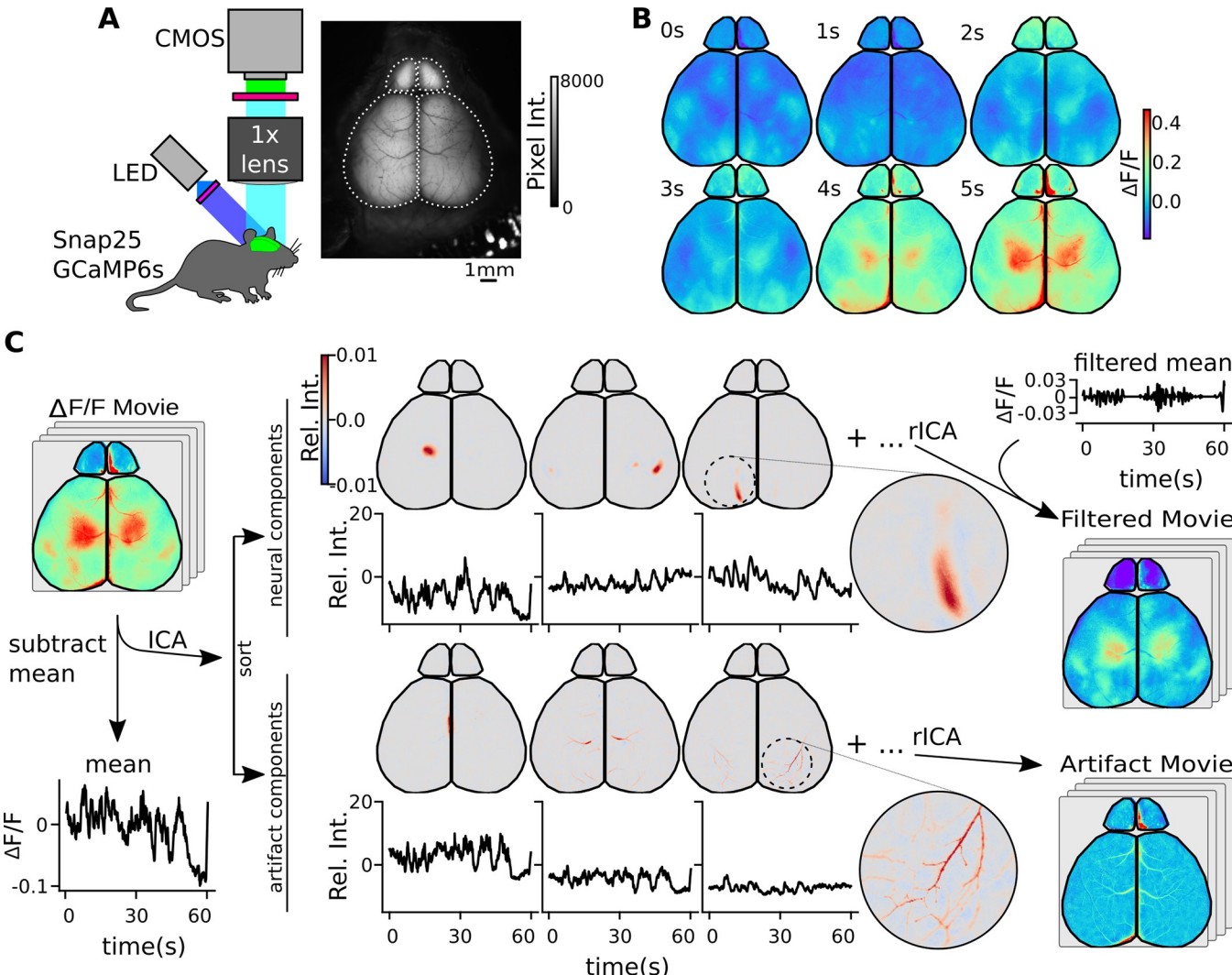

**Fig 1. ICA separates calcium data into its underlying signal components.** (A) Recording schematic and fluorescence image of transcranial calcium imaging
preparation. Full image with masked regions of interest (ROI, dashed lines) shows regions used in ICA decomposition. (B) Sample video montage of raw video
frames after ΔF/F calculation. (C) ICA video decomposition workflow. The ΔF/F movie without the mean is decomposed into a series of statistically
independent components that are either neural, artifact, or noise associated (not displayed). Each component has an associated time course from the ICA
mixing matrix. Neural components can be rebuilt into a filtered movie (rICA). Alternatively, artifact components can be rebuilt into an artifact movie. Circular
panels show higher resolution spatial structure in example in the rightmost components.

hemodynamic responses, similar data was recorded and processed in three sets of age matched control mice: cx3cr1 GFP (microglia; mGFP), adhl1 GFP (astrocyte; aGFP), and the non-transgenic C57black/6 (Bl6) mice.

## ICA separates signal sources from high resolution data

A spatial ICA decomposition on a video, wherein the global mean was subtracted, produces a series of spatial independent components (sIC) and a mixing matrix, which captures the component's influence at each frame in the video (Fig 1C). The components are sorted by temporal variance and oriented so that they all represent positive spatial effects (See Methods). The independent components can be sorted into 3 major categories based on their spatiotemporal properties: neural components, artifact components, and noise components (not shown). Group analysis on these components was performed and discussed later in this manuscript.

Neural components represent a distinct area of cortical tissue, which we refer to as its cortical domain. The spatial morphology of these neural components can vary in both spatial extent and eccentricity. Neural components are also frequently found to contain multiple domains, which have similar enough activation patterns to be identified as a single neural component. In the examples in Fig 1C, the second neural component appears to represent a secondary somatosensory network, with multiple domains on the right hemisphere, and a small mirrored domain on the left hemisphere.

Artifact components can take many forms, including those from blood vessels, movement, and optical distortions on the imaging surface. The left two artifact examples (Fig 1C) likely represent hemodynamics from the superior sagittal sinus vein (left, center) and blood flow through the middle cerebral artery (right) [33]. A very high-resolution map of the vessel patterns can be rebuilt from these components, with branching structures as small as 12 μm in diameter (shown in Fig 1C, right). Noise components lack a spatial domain and have little to no temporal structure. Signal and artifact components can be sorted manually in the provided graphical user interface (S1 Fig) or with a machine learning classifier.

Video data can be reconstructed using any combination of these components. In particular, a filtered video can be constructed by excluding all artifact components. The artifact movie can also be reconstructed to verify that desired signal was not removed with the artifact filtration (S1 Video).

## Recording high resolution data improves noise separation and increasing resting state data length results in a stable number of signal components

Neural and artifact components can be separated from noise components based on their spatial structure, log temporal variances, or lag-1 autocorrelations. Both neural and artifact components have discrete spatial structure and a high lag-1 autocorrelation. Conversely, noise components have dispersed signal across the cortex and have a low lag-1 autocorrelation. The lag-1 autocorrelation metrics from these two groups are discrete enough that these populations can be separated by their lag-1 autocorrelation alone (Fig 2A). Examples of both neural (n) and artifact components are shown, and further distinction of artifact components were further distinguished as vascular (v) and other (o) for descriptive purposes.

To automate this sorting process, a two-peak kernel density estimator (KDE) was fit to the histogram of lag-1 autocorrelation data. The KDE distribution is an easy way to summarize the two major peaks, as well as the minima between them, which defines the noise cutoff. The locations of these peaks, and the minima between them is highly stable across our 8 test recordings at P21. We found the non-noise peak (top) at an autocorrelation of $0.94 \pm 0.01$, and a noise peak (bottom) at $0.13\pm0.01$. The central cutoff minima was slightly more variable, with

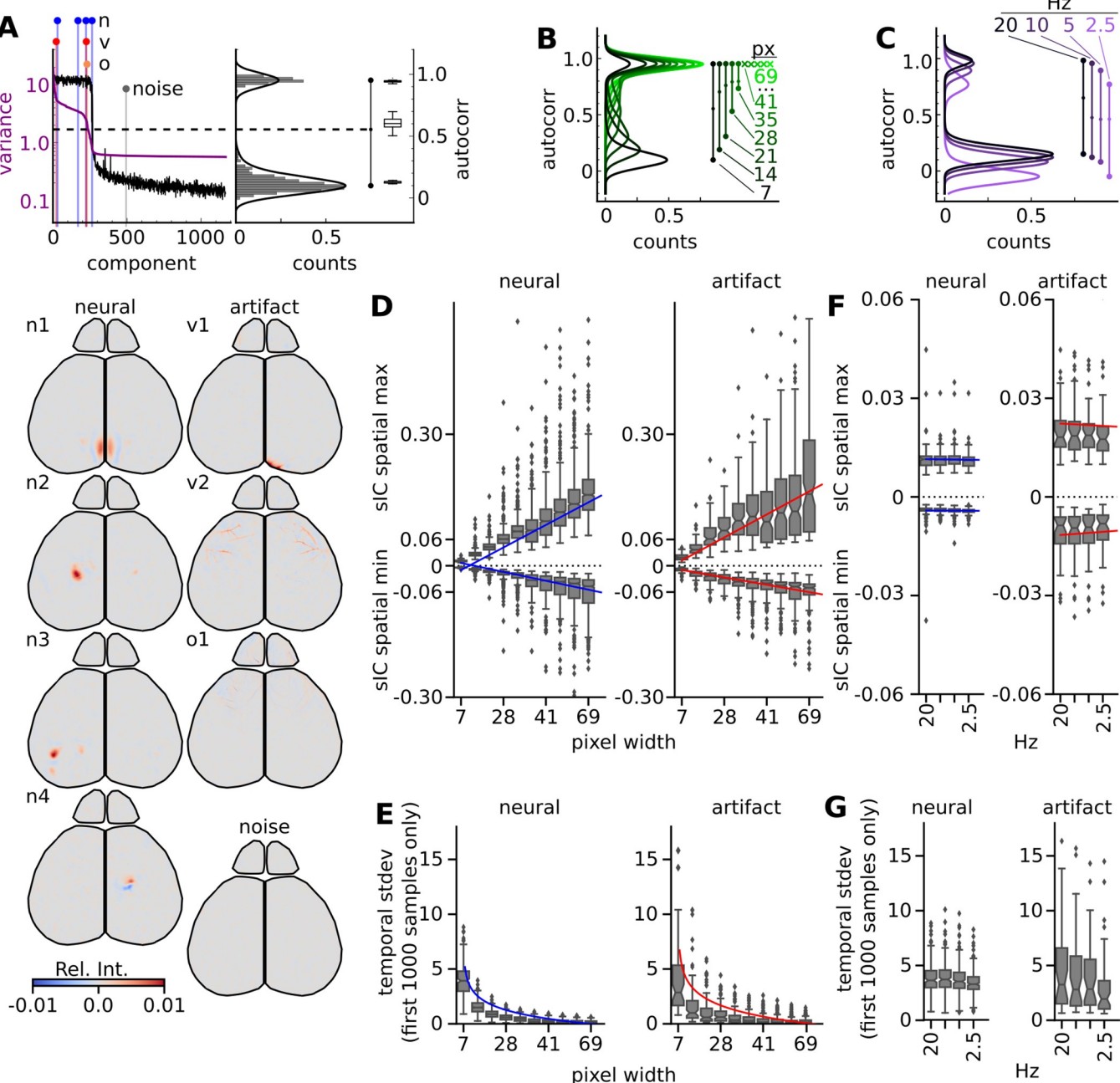

**Fig 2. ICA decomposition quality is sensitive to recording spatial and temporal resolution.** (A) Distributions for lag-1 autocorrelation (black) and log temporal variance (purple) are displayed for components 1–1200. A dotted line represents the cutoff determined from the distribution in the right panel. In the right panel, a horizontal histogram on the lag-1 autocorrelation with a two-peaked kernel density estimator (KDE) fit reveals a two peaked-histogram, summarized by a barbell line. Group data for each peak, as well as the central cutoff value is summarized by the boxplots on the right (n = 16 videos; from 8 different animals). Examples of neuronal (n; blue), artifact (vascular—v; red/other -o; orange), and noise (gray) ICs are indicated in the variance plot are shown below. Comparative ICs across spatial down sampling are shown in S2 Fig and temporal down sampling show in S3 Fig. (B) 2-peaked KDE fits of horizontal histogram distributions under various spatial down sampling conditions, with barbell summary lines on the right. After spatial resolution decreases beyond 41 μm pixel width (px), this two peak structure collapses, and an x denotes the primary histogram peak. (C) 2-peaked KDE fits of horizontal histogram distributions under four temporal down sampling conditions, with barbell summary lines on the right. sIC global max and min of each spatially (D) and temporally (F) down sampled experiment and their temporal standard deviation of the first 1000 data points (E,G).

an autocorrelation value of 0.61±0.05. A high degree of separation between these peaks (dp−p = 0.82±0.01; p < 0.001) suggests that the signal and noise signal sources were completely separated, and thus all signal sources were distinctly identified.

To test how ICA component separation is affected by spatiotemporal resolution and video duration, we altered properties of the input video and observed its effects on the quality of signal separation through lag-1 autocorrelation distributions and assessed the resulting components. Reducing the spatial resolution resulted in a steady decrease in peak separation, until the dual peaked structure collapsed at a resolution of 41 μm pixel width (Fig 2B). Increasing the sampling rate above 10Hz showed little to no effect on the peak-to-peak distance (Fig 2C; Δp−p < 0.01), and a slight decrease in the autocorrelation of the primary peak (Δp1 = 0.03). However, temporal down sampling below 10Hz resulted in a shift of the signal and noise peaks (Δp1 = 0.06), and a reduction in the peak-to-peak distance (Δp−p = 0.02). This result agrees with previous analyses that found 10Hz to be the minimal sampling frequency required for measuring population calcium dynamics [15].

We further assessed the components of both the spatial (S2A Fig) and temporal (S3 Fig) down sampling and found they continued to separate neural and artifact signals. However, the spatial down sampling experiments showed a drastic increase in the number of components necessary to capture the signals (increasing peak height of the top peak Figs 2B and S2B). In addition, the spatial maxima increased at a rate of 0.017 relative units and minima decreased at a rate of 0.006 for each increased binning (Fig 2D). The tradeoff of this increase in spatial representation, both in terms of relative numeric representation and total number of components, resulted in an exponential decay at a rate of 1.15 of the temporal features of neural components, and a rate of 1.48 for artifact components (Fig 2E). Temporal down sampling had little to no effect on the number of components (peak height of the top peak Fig 2C), the maxima and minima of the spatial components (Fig 2F, rate change: -0.0001) or on the temporal variance (Fig 2G). While the temporal standard deviations are not directly comparable in this analysis, as they represent different time courses, we included the statistics for completeness of story.

These findings suggest that the separation quality of captured dynamics are highly sensitive to spatial resolution but not as sensitive to temporal resolution. We considered collecting spatial samples higher than our current resolution of ∼6.9 μm/px, but computing decompositions on datasets this large would push the limits of available computing.

To determine the duration of video that would maximally segment the cortex, we calculated the number of significant neural and artifact components for decompositions at various video durations (Fig 3A). We found that for ICA decompositions on activity patterns from a P21 mouse, the number of neural components leveled off after 1200 seconds/20 minutes. Population analyses showed that this number was highly similar among P21 mice (n neural components: 244 ± 25.7; n artifact components: 87.2 ± 20.7; N = 3 mice, 2 subsequent recordings each).

Increasing decomposition times resulted in the temporal representation exponential decay of 0.717 per 100 seconds of raw data (Fig 3B). However, unlike the spatial down sampling, the standard deviation remained well above 0, with the standard deviation levels off similar to the number of components. The neural sIC maxima value continued to increase with more time, while the minima stayed constant (Fig 3C). In contrast, the maxima increased and the minima decreased for the artifact sIC. Examples of comparable neural and artifact sIC with varying amounts of data included in the decomposition are shown (Fig 3D). This shows that more data included in the decomposition results in a more spatially restricted, temporally rich sIC.

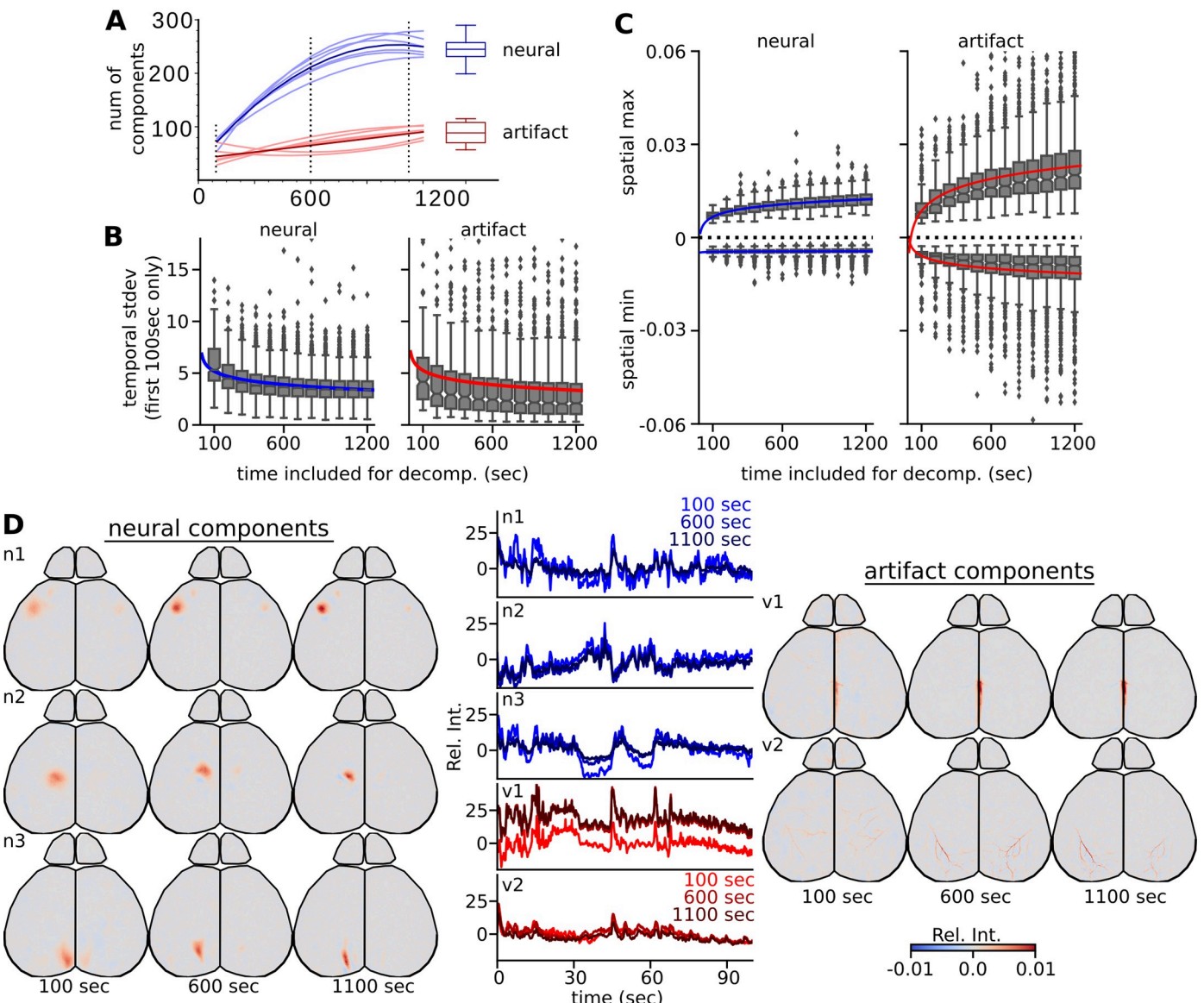

**Fig 3. Increased duration increases specificity of each IC.** (A) Component stabilization for different length video subsets of six 20- minute video samples. (n = 6 videos from 3 different animals) Individual thin lines show polynomial fit to neural or artifact components under each time condition. Thick lines denote the curve fit of the mean number of components in each category across these six experiments. The group distribution of components at 20 minutes is summarized by the boxplot on the right (n = 16 videos; from 8 different animals). Dashed vertical lines are durations in which examples are shown in D. (B) Temporal standard deviation of the first 1000 data points. (C) sIC global max and min of each varying duration included in each decomposition. (D) Examples of similar ICs between decompositions of varying lengths for both neural components (left sIC with central blue time courses) and artifacts (right sIC with central red time courses).

## Spatiotemporal metrics can be derived from each component to assess the classification of each signal source

Using the ICA decomposition from 20-minute duration videos, we inspected each set of experimental components from both controls and GCaMP6 expressing mice to classify each as neural or artifact (Fig 4A). Neural components typically have globular spatial representation with highly dynamic properties. Vascular artifact components can be easily visually identified by the vascular-like spatial representation. Other artifact components that are commonly seen in the components are movement or preparation artifacts. These typically have a diffuse spatial

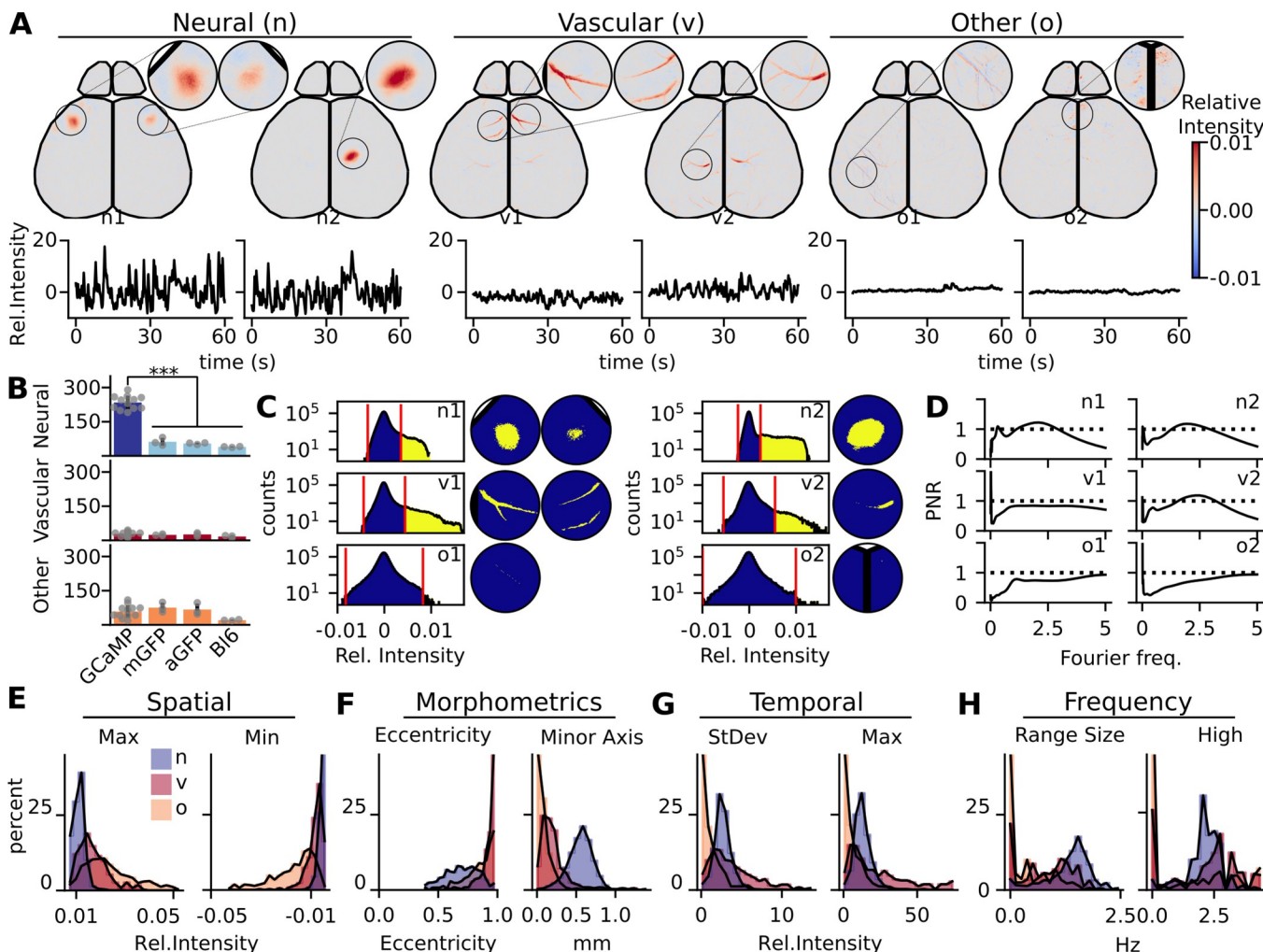

**Fig 4. Class identity cannot be established by any individual extracted feature.** (A) Examples of independent components of neural (n) signal, vascular (v) artifacts, and other (o) artifacts. Components are defined by both the sIC and its temporal fluctuations. Circular windows magnify key portions of the sIC. sIC values represented by colormap from blue to red. Temporal representation is in relative intensity (black time course under the sIC), only 1 minute of the full 20 minutes are shown. (B) A comparison of the number of neural signal (GCaMP: dark blue; controls: light blue) and the artifact components (vascular: red; other: orange) with each animal shown (GCaMP components: N = 12 animals, n = 3851; mGFP components: N = 3, n = 484; aGFP components: N = 3, n = 442; WT components: N = 3, n = 229). (C) Examples of binarization of the sIC. Histogram shows the full distribution of sIC values. The dynamic threshold method to generate binarized masks was used to identify the high sIC signal pixels (yellow) against the gaussian background (blue). Windowed spatial representation shows binarization on the key portions of the sIC. (D) Examples of neural and artifact wavelet analysis shown in the power signal-to-noise ratio (PNR) plots. 95% red-noise cutoff was used to create signal to noise ratio (black dashed lines). (E) Histograms of example spatial metrics derived from GCaMP sIC values, (F) morphometrics from the shape of the binarized primary region, (G) temporal metrics derived from relative temporal intensities, (H) frequency metrics derived from the PNR.

representation with smaller or sparse temporal activations. We manually scored each component in the dataset as an artifact (vascular or other) or neural component (Fig 4B). From all the GCaMP experiments, an average of 73.5 ± 5.9% of the components were identified as neural, where the remaining 26.5±6.3% were artifact (vascular: 8.7 ± 2.7%; other: 17.6 ± 7.1%). GCaMP mice had substantially higher numbers of neural components compared to the controls, resulting in four times as many as the GFP mouse lines and six times the number in Bl6 mice (mean number of neural components GCaMP: 235, mGFP:62, aGFP:54, Bl6: 39).

We extracted spatial and morphological metrics of the neural and artifact components to characterize spatial feature differences (Fig 4C). We can pull general spatial intensity metrics

like global minimum and maximum from the spatial sIC of each component. The largest sIC values correspond to the regions that have the most dynamic change from the data. Given that the shapes of the high pixel intensity values are used by humans to identify their classification, we decided on a dynamic thresholding technique to binarize the sIC. When examining the histogram of intensity values of the neural sIC, there is a large population of pixels centered around zero with a single long tail. We identified all pixels that were unique to the long tail by excluding all values that lie within range of the shorter gaussian tail. From these binarized masks, morphometrics of each primary region of the component can then be quantified, such as the axis lengths or eccentricity of the shape.

We characterized temporal dynamics of each component by extracting features from the component time series (Fig 4D). The time series analysis allows us to pull out temporal features of each component, such as standard deviation and global maxima/minima of each component contribution. We performed wavelet analysis on these time series to characterize only highly significant frequencies (S4 Fig). We calculated a power signal-to-noise ratio (PNR) with the 95% quantile of red noise defined by the autocorrelation value of each time series. With this ratio, significant frequencies resulted in a value above 1.

After extracting these metrics, we then compared the diverse populations of neural and artifact components, separated between vascular and other, for each feature of interest (Fig 4E–4H). A full list of all metrics and their respective definitions is provided, organizing between spatial, morphometric, temporal, and frequency features (S5 Fig). While the data shows general trends, there is not one single metric that alone could predict the classification of artifacts and neural components.

Control neural components are not distinct globular regions like those from the GCaMP line (S6 Fig); rather, they had co-activity with vascular units in the center of its domain. This resulted in the thresholded region being more similar to the vascular artifacts seen in GCaMP components. These identified neural components did not seem to follow known neuronal functional or cytoarchitecture, but were primarily a result of localized vascular and hemodynamic influence. However, we were still able to find example components that only had vascular spatial representation without the surrounding tissue activation. Finally, we found similar artifacts of the other category in the control data that are also present in the GCaMP sets of independent components.

## GCaMP mice have strong distinct globular domains that cover the entire cortical surface

To investigate how well these metrics captured features of each component, we explored the coverage of the cortical surface with regions identified by the dynamic thresholding technique. By plotting all the contours from one experiment, the major footprint of the component shows its representative space within the brain region (Fig 5A). The majority of defined brain regions are represented by the GCaMP component footprints, with varying amounts of overlap associated with different cortical areas. Control data resulted in sparsely mapped footprints across the cortex. Further, the mapped centroid location of all components that had a thresholded region from all GCaMP experiments shows the neural components have high densities in the sensory regions of the brain (Fig 5B).

Thresholded GCaMP neural components have high densities in the olfactory bulbs and posterolateral portions of the cortex, including visual, auditory, and somatosensory systems. There is less dense localization of centroids along the anteromedial portions of the cortex, including motor and retrosplenial cortices. Further, in both the GCaMP and control mice, we see the majority of artifact components localize along anatomical brain vasculature. The major

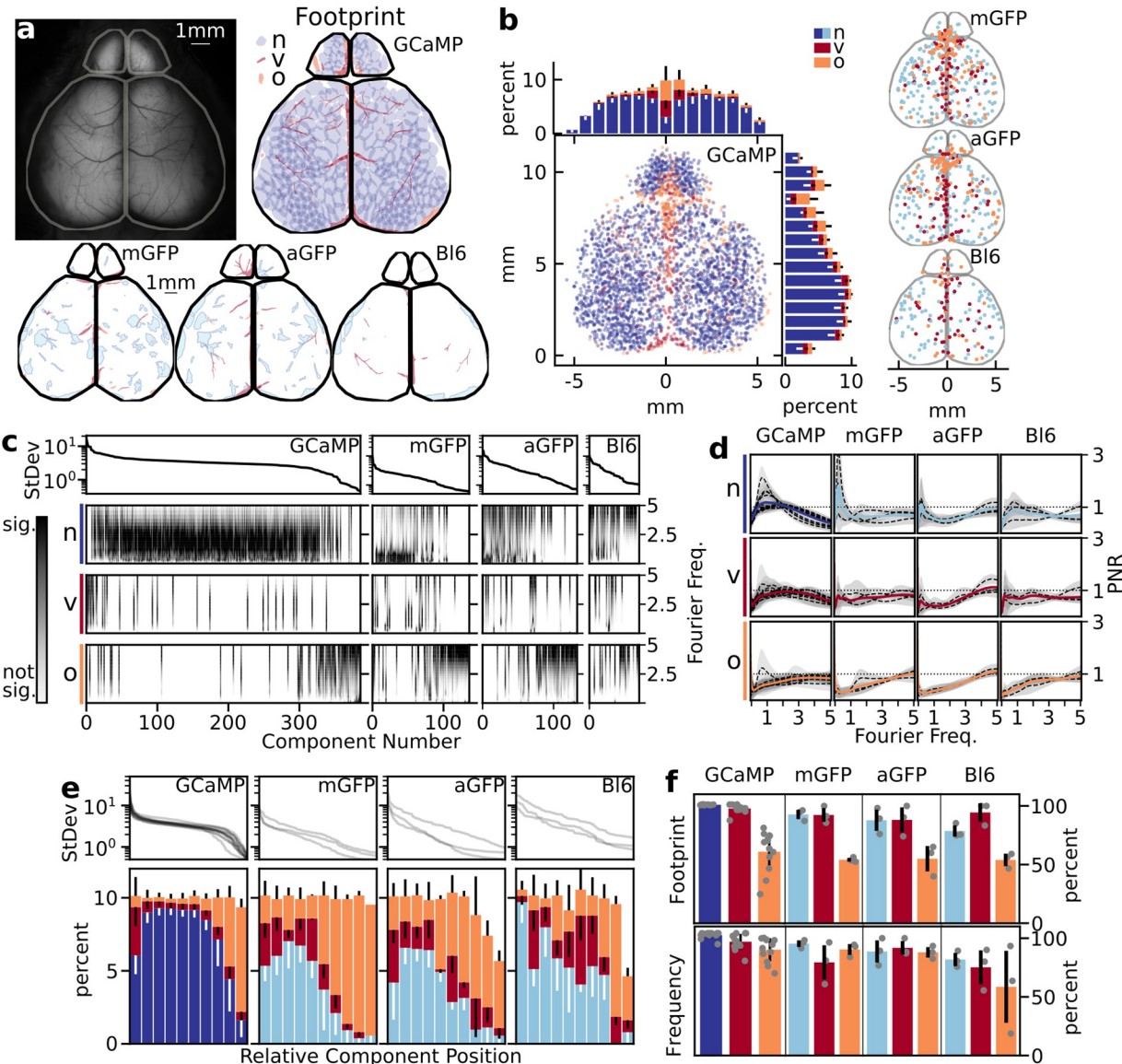

**Fig 5. Spatial thresholding and frequency data reliably produce neural metrics.** (A) Individual experiment preparation with corresponding spatial footprints by class of component: GCaMP neural (dark blue), control neural (light blue), vascular (red), other (orange). (B) All model experiments (N = 12) with corresponding centroid location of each of the class of metrics. Histograms show the resulting average distribution of spatial location across the field of view (error bars are standard deviation between experiments). (C) Individual experiment (same as A), where components are sorted by temporal variance. PNR mapped to each component and organized between the classification of components. (D) Main frequencies seen in each component class between each experimental condition. Dotted lines represent the mean dominant frequency within each animal, where the gray around the mean corresponds to the standard deviation of that animal. The color line corresponds to the grand mean between all experiments. (E) Relative position-based variance of the types of components between experiments and transgenic model, shown as the average and standard deviation between experiments. (F) The percent of components that had footprints and frequency data that was above the noise cut-off, separated by component type and experimental condition.

venous systems, including the rostral rhinal vein, the superior sagittal sinus, and the transverse sinus, all show high densities of artifact centroid locations [33]. The cerebral arteries are less consistent in localizing the primary domain of their respective components. We see that many of the other artifacts align with the sagittal and lambda cranial sutures [34].

We investigated the effects of wavelet analysis on feature generation by sorting each component within an experiment to its temporal standard deviation value. We then ordered each class of components based on variance and displayed a grayscale heatmap of the significant frequencies in each component across experimental conditions (Fig 5C). Taking the average of each of the global wavelet spectrum across each experiment highlighted the prominent frequencies seen in each classification (Fig 5D). Prominent GCaMP frequencies are between 0.3 to 3.5 Hz, where control dynamics are typically seen between 0-1 Hz. Vascular components tend to have the same frequencies as their neural counterparts, while other components typically have faster frequencies (above 3 Hz), most likely due to motion during the recordings.

We looked at the overall distribution of the class of components (neural, vascular, other) with respect to their relative variance (Fig 5E). We found significant shifts in distribution in the types of components based on variance. Neural components were found with high variance, however they significantly tapered off nearing the noise floor. We found the highest percentage of vascular components with high variance, followed by constant low probability throughout the relative variance. The other components had the highest probability close to the noise floor, with low frequency throughout the rest of the relative variance position.

Among all GCaMP experiments, 92.3 ± 5.6% of the components had a strong spatial activation, resulting in a thresholded region to assess. However, when we looked at the breakdown of the class of components, we found that 99.8 ± 0.2% of all neural components had a thresholded region (Fig 5F). The artifacts had fewer thresholded components, specifically in the other classification; vascular artifacts had 96.6 ± 3.6% and other artifacts had 60.2 ± 16.2% within their respective class of components that had a domain footprint above the spatial threshold. Of note, we found that components with low variance close to the noise threshold had increased probability of not having a domain threshold.

Wavelet analysis between all GCaMP experiments revealed 95.8 ± 3.8% of the components had statistically significant frequencies to assess. We found that 98.4 ± 2.5% of all GCaMP neural components had significant frequencies (Fig 5F). Vascular artifacts had 93.0 ± 6.5% and other artifacts had 86.6 ± 9.4% of components with significant wavelet frequencies within their respective class of components.

Overall, this indicated that all metrics can be generated for the vast majority of neural signals and vascular artifacts. However, on average, 40% of the other artifacts do not have morphometrics for their components and 13% are lacking significant frequencies. Further, this also shows changes in the relative spatial and temporal variance distributions of each of these components. The neural and vascular components align with known and predictive anatomy and were found primarily in mapped locations of increased variance. The majority other components have less variance, and therefore less contribution to the original dataset; the ones that had a footprint were typically found along cranial sutures. All control data had fewer footprint and frequency metrics (Fig 5A and 5C).

## Spatial metrics best separate neural components from artifacts

To build a classifier, we identified metrics that distinguish between neural and artifact components. Correlation of component class to with respect to their metric value and their respective t-statistic between class identified which features are most useful to classify each component (Fig 6A, top). For this process, we randomly selected seven animals from our twelve-experiment dataset for determining features for training the classifier. The remaining five were used as the novel dataset for validating the machine learning performance.

We trained the random forest classifier with all metrics to identify the importance of each feature (Fig 6A, middle). The features with the greatest t-statistic magnitude had the highest

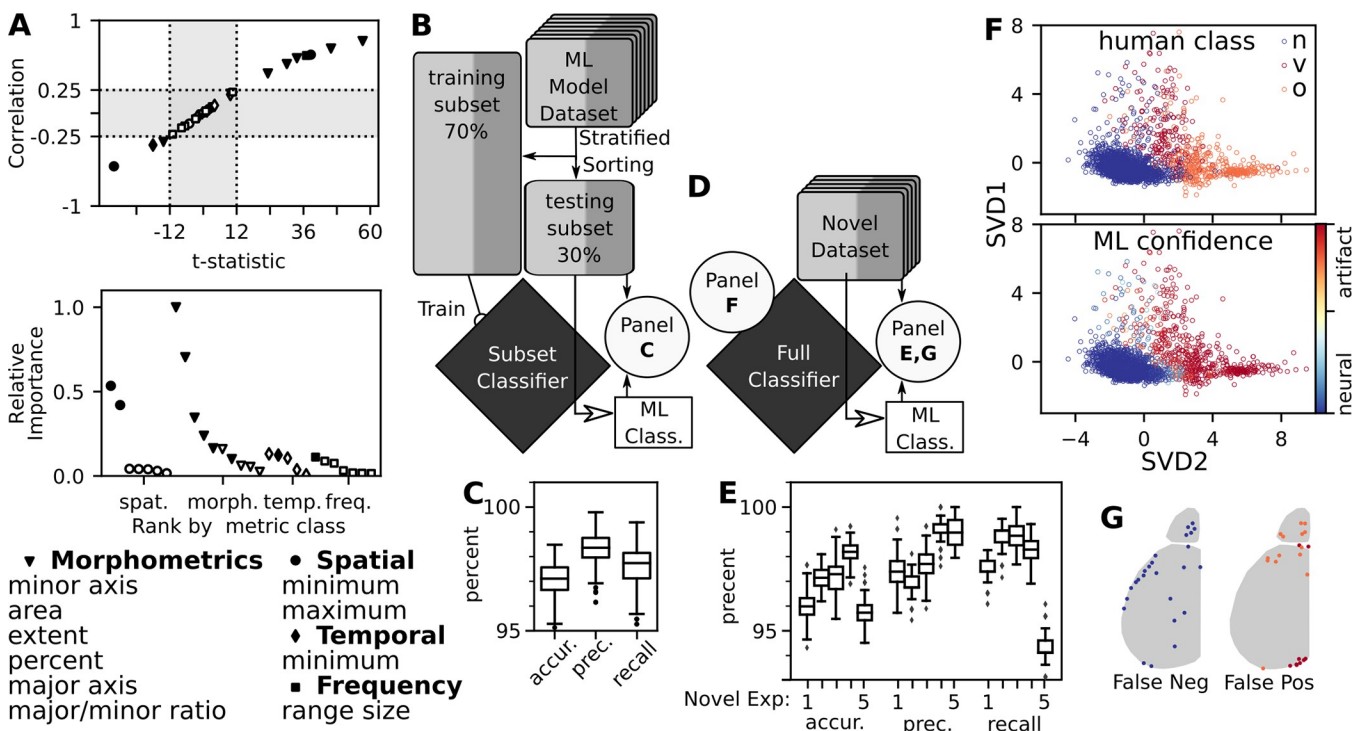

**Fig 6. Spatial and morphological metrics are most important to classify components.** (A) Correlation and t-statistic between artifact and neural components for each feature (N = 7, n = 2190). Spatial (circles), morphological (triangle), temporal (diamond), and frequency (square) metrics plotted. Cut off values that helped in the selection process are dotted lines, rejected values in gray. Closed points are components that meet requirements. Relative importance metric from the Random Forest classifier plotted against each metric by their respective classes. Selected metrics shown in the list within each type of feature, sorted by greatest t-statistics magnitude. (B) The dataset was parsed into ML modeling dataset (N = 7, n = 2190) that was used to establish the machine learning pipeline and a novel dataset (N = 5, n = 1661) of full experiments that will not influence the classifier. Modeling data was stratified 70/30 split based on classification. 1000 iterations of training the machine learning classifier on selected metrics and validating the machine classification with human classifications. (C) Performance of the ML training, using subsets of the ML modeling dataset. 1000 iterations resulted in accuracy, precision and recall boxplots. (D) 1000 iterations of training on the full ML building dataset was performed and the novel dataset was assessed on its performance. (F) SVD projection of metric data with human classification mapping (top) and the confidence of the ML classifier (bottom). (E) Performance of the classifier on each of the novel datasets, animals plotted separately showing distribution of the 1000 different trained classifiers. (G) Approximate location of false negatives and positives from novel datasets.

importance for proper classification. In particular, spatial and morphological metrics were found to have the highest relative importance for component classification. The final list of 10 feature metrics utilized in the machine learning process are shown (Fig 6A, bottom).

## Machine learning performs as well as human classification

We utilized the common approach of hiding a portion of the data from the learning algorithm to validate efficiency of machine learning and establish hyperparameters (Fig 6B). Stratified sorting was used to ensure an equal ratio of artifact and neural signal was placed into each subset. We sampled and trained the random forest classifier 1000 times to see the distribution of results and found that it performed well by all metrics assessed: mean accuracy of 97.1%, mean precision of 98.4%, and mean recall of 97.6% (Fig 6C). All other tested algorithms did similarly well (S7A Fig).

After establishing the efficacy of the classifier, we set out to assess the full classifier based on all data points from the machine learning dataset. We projected all features onto the first two components of a singular value decomposition (SVD), mapping both the human classification and the mean classifier confidence for 1000 iterations (Fig 6F). As expected, we saw distinct

neural and artifact clusters in feature space. Interestingly, the two different types of artifacts also separated into distinct portions of the projected feature space. The confidence of the classifier showed very few components between the extremes, illustrated by the top binned confidence value distributions for each human classification (S7B Fig). We found 71.2 ± 0.2% of components were binned in highly confident values for neural signals (left-most bin), and 22.7 ±0.2 were binned in highly confident values for artifacts (rightmost bin) (7.0±0.2% for vascular; 15.7±0.1% for other). This indicates that the classifier exhibits reliable confidence in the decision boundaries.

To assess the efficacy of this classifier, we then tested 1000 iterations of novel data—completely new experiments that were not involved in training the classifier (Fig 6D). We plotted the resulting 1000 iterations of each experiment separately (Fig 6E). Notably, we found that the overall results were about the same as the subset classifier: mean accuracy of 96.9%, mean precision of 98.0%, and mean recall of 97.6%. From the histogram of classification frequency, we found similar results to the confidence of the classifier (S7C Fig). Among all components, 69.7±2.0 were confidently classified by machine learning as neural signal (left-most bin), where 25.8±1.4% were confidently classified as artifact (rightmost bin; 7.5±0.4% of vascular; 18.2±1.6% of other). The remaining 5% were mis-classified. We investigated the locations of the components that consistently showed false positive or false negative (Fig 6G). The majority of these components were either on the edge of the region of interest for the cortical hemispheres or within the olfactory bulb.

## Global mean needs a high-pass filter to account for removed artifacts before re-addition

Removing artifact components will ensure that neural signals are the dominant signal after rebuilding with identified neural components; however, during reconstruction of ICA data, re-addition of the global mean must occur. Thus, we examined the influence of removing artifact components on the global mean and how filtration of the global mean should be considered. For example, vascular artifacts associated with the superior sagittal sinus contribute to the global mean and increase the range of signals recorded during periods of motion (S9 Fig). Assessment of the global mean from GFP control experiments showed pronounced signal in these slower frequency oscillations, suggesting the use of a high pass filter. Indeed, we found that application of a high-pass filter with a 0.5 Hz cutoff minimizes these types of global slow oscillations (S8 Fig). This type of filtration should not be applied to each component individually, as there are regional networks reliant on these slower oscillations [15], but this filtration approach does remove all global oscillations, some of which are neural based fluorescence (discussed below). However, removal of these low frequencies from the global mean improved identification of the cortical patch signal sources that contribute to neural activation (S2 Video).

## Domain maps optimize time course extraction from underlying data

In addition to their applications for filtering, the components also are a rich source of information about spatial distributions of signal within the cortex. Components across the cortex show a wide diversity of spatial characteristics and represent the detection of independent units of signal. We use the spatial domain footprints of each signal component to create a data-driven 'domain map' of the cortical surface by taking a maximum projection through each component layer (Fig 7A). When we compare the number of sIC (249±22) and the number of domains (261±23), we see a significant increase in the number of domains from the number of sIC (Fig 7B; paired t-test p<0.001). Due to the competitive nature of max projection and the

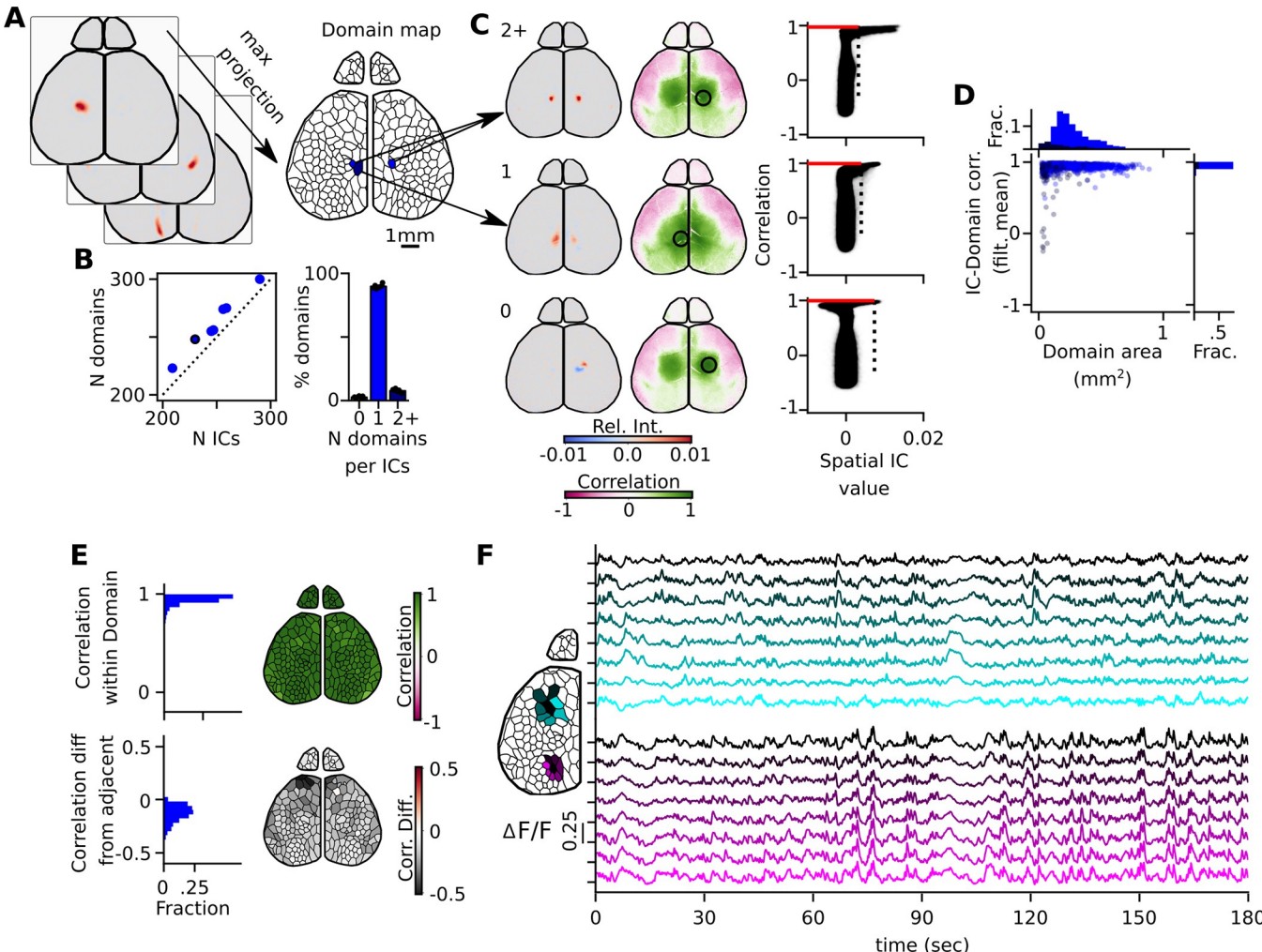

**Fig 7. Domain maps represent the features of the ICs.** (A) Schematic of domain map creation. A maximum projection is taken through each blurred neural component to form a domain map. Mean time courses are extracted from rebuilt filtered movies within a defined domain. (B) Left scatter plot compares the number of neuronal ICs and the resulting number of domains and the dashed line is the identity line. The circled dot indicates the examples used in this figure. Right bar plot shows the fraction of ICs that have major contributions to the number of domains. (C) Examples of ICs that contributed two domains, one domain or had no contribution. The location of the maxima (black circle) of each IC was found and a point correlation map of the rebuilt full resolution filtered movie was produced (green and pink maps). Pixel-wise scatter plot (right) shows the relationship between the spatial IC value and the correlation map value. Dotted line is the threshold value described in Fig 4. Red line indicates the median correlation value of the correlation map that resides in the thresholded IC. (D) IC-Domain correlation with domain map created with filtered mean plotted with respect to the size of each domain. Dark blue corresponds to domains where each IC made more than one domain. Bright blue corresponds to those that made only one. (E) Median correlation value based on point-correlation analysis within each domain (top). Difference in correlation between the center domain with all its adjacent neighbors (gray scale value corresponds to the mean of the median difference value found in each immediately adjacent domain). (F) Example time courses of domain neighborhood. Left domain map identifies the location of each neighborhood with each corresponding time series. Example ICs and point correlation data shown in S10 Fig.

frequent number of sIC that have multiple domains, we looked further into the map creation. We were able to match up each domain with its most influential sIC. The majority of sIC had great influence on only one domain (89.5±1.50%). A smaller percentage of sIC had influence on two or more domains (7.55±1.24%). Finally, an even smaller subset of sIC did not have any influence on the domain creation (3.00±0.77%).

At full resolution, there are approximately 1.7 million pixels along the surface of the cortex and olfactory bulb–an impractical number of sources for most network analyses, which work best on 10–300 time series [35]. As such, data-driven domain maps are an optimal method for extracting time courses from the cortical surface. Time series were extracted by averaging the

filtered movie under each domain. This results in a series of 261±23 time series per video recording, representing a ∼ 6,500-fold compression rate (Fig 7A; right). In this sense, we no longer have an independent component, but rather the full timeseries of the underlying neural signal. We do this to ensure that every aspect of the neural signal is included in the final analysis, especially those that did not influence the domain map.

As we have identified the sIC and its most influenced domain, we are able to make direct comparisons between the temporal dynamics of the sIC and domain. Further, to understand the relationship between sIC, domain, and the full resolution filtered dataset, we performed a pixel-wise point correlation analysis across the full resolution filtered data at the location of neural sIC maxima (Fig 7C). These three pieces of data help us make several direct comparisons to understand the sIC, its corresponding domain, and their relation to the filtered data. First, to assess the independence of each sIC, we make a direct pixel-wise comparison of the sIC value and the matching point correlation map. We see that only the most highly correlated values are associated with the sIC high values. Second, we can directly measure how each temporal feature of the sIC correlates to the rebuilt time series of its most influenced domain (Fig 7D). With the filtered mean re-added, we see that the sIC and its corresponding domain are highly correlated (Pearson's correlation coefficient: 0.93±0.08). When we identify the multiple domains that arise from the same sIC, they tend to be small domains (dark blue).

To assess how well the domain map represents the neural signal in the full resolution filtered data, we can take the median of all correlation values residing in the domain-matched point correlation map (Fig 7E; top). Within each domain, the median pixel-wise correlation coefficient to the sIC maximal location is 0.92±0.07. Further, we can assess how independent each domain is compared to its shared bordered neighborhood. To test for independence, we investigate the null hypothesis; that is, the difference between two point correlation maps associated with adjacent domains should be zero. Two such neighborhood time series (Fig 7F), their corresponding ICs, point correlation maps, and the correlation difference between surrounding and the center domain correlation maps are shown (S10 Fig). In each of the difference maps, we see shifting correlational differences even in adjacent sIC/domains. To quantify how distinct each center domain is to its corresponding surrounding domain, we take the mean of all median differences that reside in the surrounding domains (Fig 7E; bottom). On average, the surrounding domains are 0.15±0.02 less correlated than its center domain. However, this is dependent on the cortical region. Note that the light gray areas reside in primary sensory areas, showing they tend to be more similar; conversely, the darker gray higher order domains tend to be more independent to their neighborhood.

To test how well the full filtered video was represented in these time series, we rebuilt 'mosaic movies', where each domain is represented by its mean extracted signal at any given time point (Fig 8A and S3 Video). By comparing the borders of the large higher order visual activation, one can appreciate that visually, the data appears more distorted in the Voronoi and grid. To numerically compare whether this method of time course extraction was superior to alternate methods, we calculated the residuals between the two movies. We also compared residuals from mosaic movies rebuilt with either grid or randomly generated Voronoi maps.

The residuals between the mosaic movies and the filtered movies were compared to the total spatial variation in the filtered movie to quantify the amount of total signal represented by the extracted time courses (Fig 8B, left). In nearly every experiment, the optimized domain map performed better than any other time course extraction method, and accounted for 68±1.2% of the total spatial signal in the filtered video (n = 8). Domain maps generated from different videos from the same animal performed nearly as well as the optimized domain maps created from the compared video (Fig 8B, right). These maps performed significantly better (p = 0.01) than the grid maps, and much better than the Voronoi maps (p < 0.001).

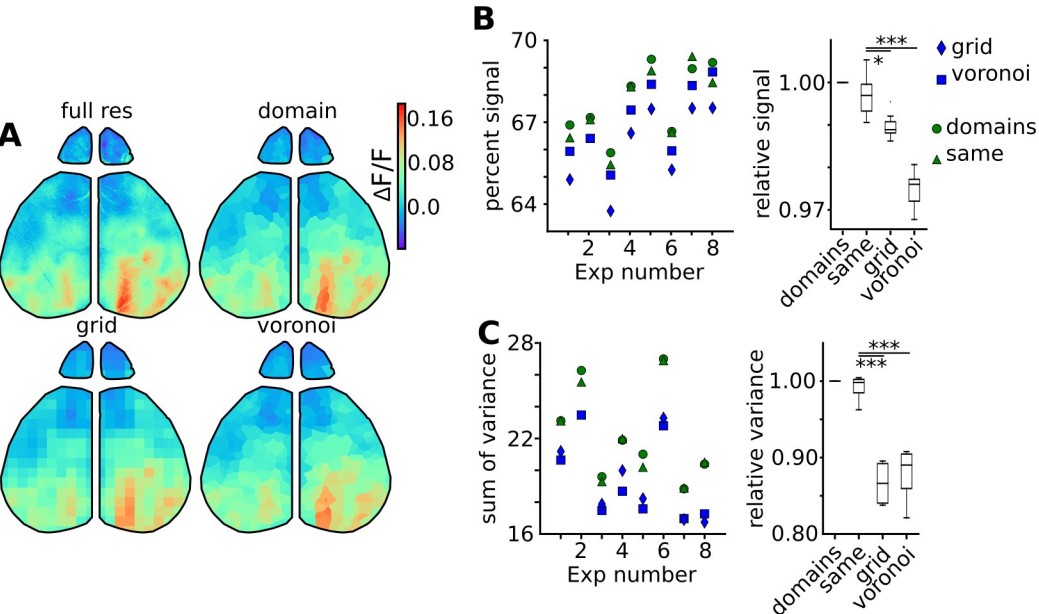

**Fig 8. Time series extracted from domain maps outperform time series generated from other down sampling methods.** (A) Example of a mosaic movie frame rebuilt with respect to each down sampling technique. The non-down sampled filtered movie is represented on the left with subsequent down sampling based on domain, grid or Voronoi maps. (B) Percent total signal of the filtered video represented by extracted time courses. Percent of overall video signal captured in domain maps was calculated for each animal (green circle; N = 8), and compared to signal content from a domain map generated from a separate video from the same animal (green triangle). Percent total signal represented by time courses extracted from grid (blue square) or randomly generated (blue diamond) maps were compared as controls. In the right panel, the percent signal relative to the domain map percent signal was summarized in a box plot. (C) Variation between time courses extracted with each map method was then quantified as a sum signal variation for each experiment. In the right panel, the sum signal variation for each comparison map relative to the optimized domain map sum signal variation was summarized in a box plot.

Saving these extracted time courses and all associated metadata results in a file size of $\sim$ 100MB, representing an additional $\sim$ 60-fold compression compared to saving the full ICA compressed dataset. One potential benefit to accounting for the underlying regions of the brain while extracting time courses is reducing the amount of times that an extracted mean signal is diluted by signal from a neighboring region. Properly restricting time series extraction to statistically independent units should enhance the dynamic range between extracted time series.

To test whether domain maps extracted time courses better extract the full range of variation in the cortical surface, we compared the total variation between time courses rebuilt under domain maps from the same video, same animal, or control grid and voronoi maps (Fig 8C, left). When normalized to the performance of the optimized domain map, domain maps from the same animal again had similar performance, but grid and voronoi maps performed significantly worse (p < 0.001; Fig 8C, right). There is a $\sim$ 15% reduction in signal variation in grid or voronoi maps compared to domain map extracted time courses.

### Animal specific domain maps can be regionalized based on reference maps and domain features

We were impressed with how the domain shape and structure seemed to capture the functionality of each patch of cortex, so we wanted to investigate the interpretations of each domain shape. We assumed that each sIC would capture underlying functional units established by

known mesoscale circuits. As such, we were curious about two key aspects that the domain shape captured underlying circuit structure: domains will have regional, non-uniform restrictions similar to established maps, and subsequent recordings would reliably produce similar domain shapes.

To investigate the domain shapes across the cortex, we calculated the same morphometrics used in our machine learning to identify neural sIC for each domain. When we looked for regional differences in domain size across the cortex, we saw a trend for large domains to reside in the anterior and medial portions of the cortex (Fig 9A, left). More impressively, when we plotted the major axis of highly eccentric domains (Fig 9A, right; top quantile of each experiment), we saw regional borders similar to the reference map (Fig 9B). These highly eccentric domains reside by defined structures such as the edge of the cortex, as well as the regions between primary cortical areas. It was not surprising that the eccentric domains reside by the physical structures, but it was surprising we were able to get consistency between individual animals in the locations of the intracortical highly eccentric domains. This led us to think there is a physical structure or circuits that influences these domains. Each domain was first manually sorted into cortical regions based on these shape observations. Second, we looked at the domain correlation maps associated with each cortex. We noted that the frontal cortex tended to have a lateral and median correlation pattern (S10 Fig), allowing us to distinguish between the medial vs lateral motor region (Fig 9C and 9D). Domains did not exhibit uniform spatial characteristics across the neocortex, as seen when we compare the regional differences in spatial characteristics, such as area (ANOVA F = 139, p < 0.001), as well as eccentricity (ANOVA F = 60.0, p < 0.001). Further, we see a significant trend in that higher order and motor regions (R, V+, Ss, Mm, Ml) had larger (p>|t| = 0.000) and more eccentric (p>|t| = 0.000) domains than primary sensory areas (V1, A, Sc, Sb, S).

To test the meaning of these maps, a series of comparisons were performed. Pairs of maps were overlaid on top of each other (Fig 9E–9G), and every domain was compared to its nearest domain in the comparison map. The Jaccard overlap was calculated for each of these domain pairs, and quantified for each pair of map comparisons. For a null hypothesis, randomly generated Voronoi maps were also compared.

Maps generated from different recordings from the same animal were found to be highly overlapping, and hence more similar (Fig 9F; p < 0.001). There was no significant difference in comparisons between littermates vs non littermates. Non-littermate map comparisons were significantly more similar to each other than to voronoi maps (p < 0.001).

We additionally quantified whether detected regions were similar across map comparisons. We again found that comparisons between maps from the same animal were highly similar (Fig 9G; p < 0.001), no difference was found between littermates and non-littermates, and comparisons between different animals were significantly more similar than a comparison between a region map and a randomly generated voronoi map (p < 0.001). For comparison, domain maps were created for all analysis done in this manuscript (S11 Fig) and can be compared with domain maps from the down sampling and duration experiments (S12 Fig). In summary, regions and domains are similar between recordings either in the same or on different animals, compared to a null map distribution.

## Validation of domain map using functional activity during locomotion

To verify our domain maps, we performed network analysis during locomotion to compare our maps with known regional changes established in the literature. We extracted time courses when locomotion and whisking occurred from the behavioral videos (Fig 10A). As reported, we see whisking occurring at the same time as locomotion [37,38], where locomotion occurred

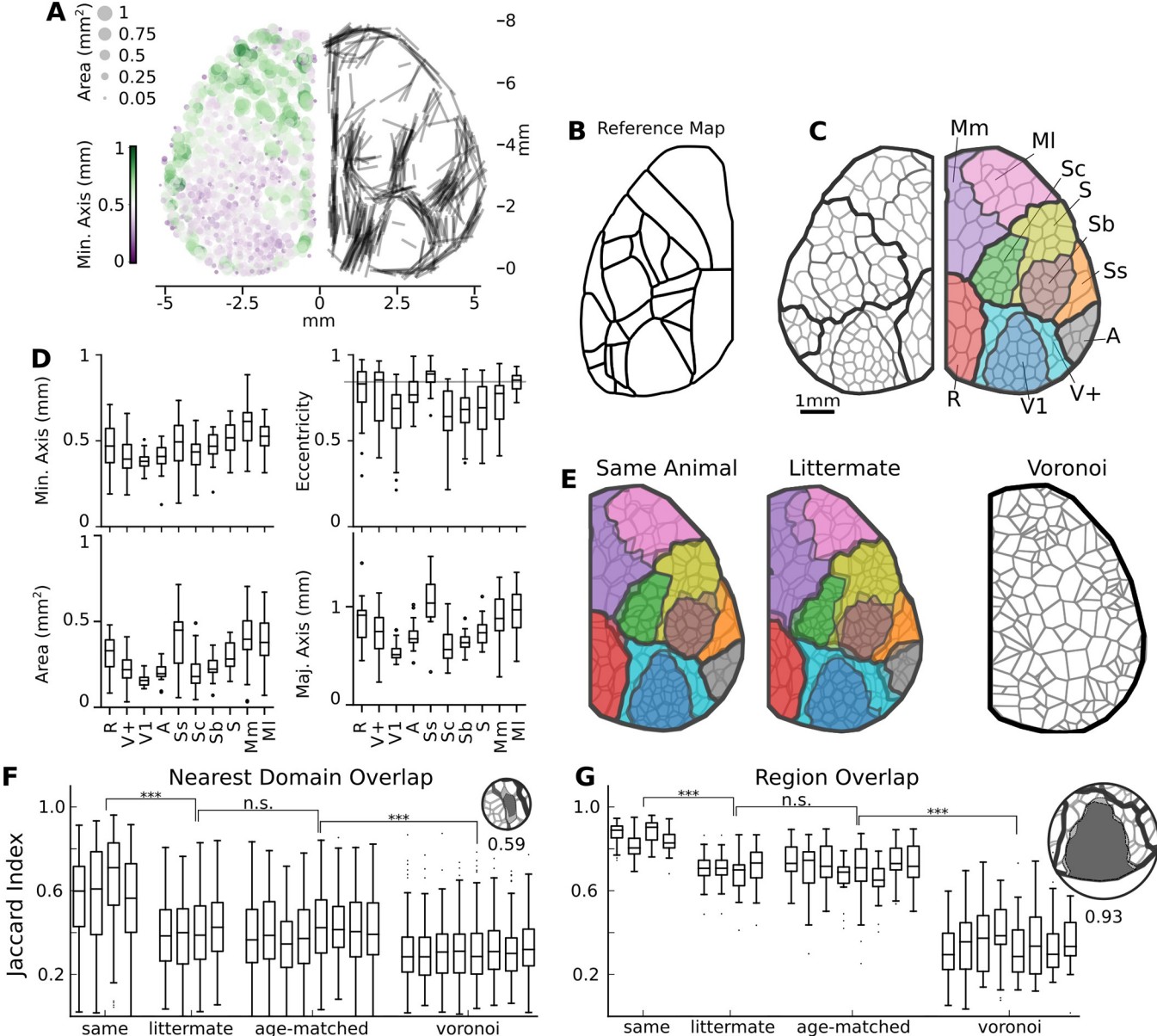

**Fig 9. Domain maps are created from ICA components and are unique to each recording, but highly similar among individual animals.** (A) Domain size (circle size) and minor axis (green-purple colorbar) plotted at each position across all hemispheres independently (left). Highest eccentric domains (top quartile) plotted showing the location and direction of its major axis (right; n hemispheres = 14, 7 mice) (B) The Allen Brain atlas map [36] is additionally used for anatomical reference. (C) The final manually assigned region, with associated labels. (D) Domain minor axis and eccentricity by region (grey horizontal line denotes top quantile used in A). 7 mice. (E) Example overlay of one domain map on another from the same animal. Individual domain or region overlap is calculated using the Jaccard index (intersect / union). Population analysis of the Jaccard index for domain (F) and region (G) overlap comparisons. Maps are generated from a different recording on the same animal, a littermate, a non-littermate, or a randomly generated voronoi map. Significance is calculated using a two-way ANOVA, followed by post-hoc t-test analysis with Holm-Sidak correction. Retrosplenial: R; V1: Visual, Higher order visual: V+; Auditory: A; Somatosensory Secondary: Ss; Somatosensory Core: Sc; Somatosensory Barrel: Sb; Somatosensory other: S; Motor medial: Mm; Motor lateral: Ml.

during 8.8±2.5% and whisking occurred 10.5±4.6% of the time assessed (Fig 10B). We first looked at the domain correlation to locomotion, comparing both the original and filtered mean (Fig 10B and 10C). The original mean is correlated to locomotion and high pass filtration of the mean eliminates this global correlation. Several studies have established that during the state of locomotion, there is a non-uniform increase of activity across the cortex [39,40],

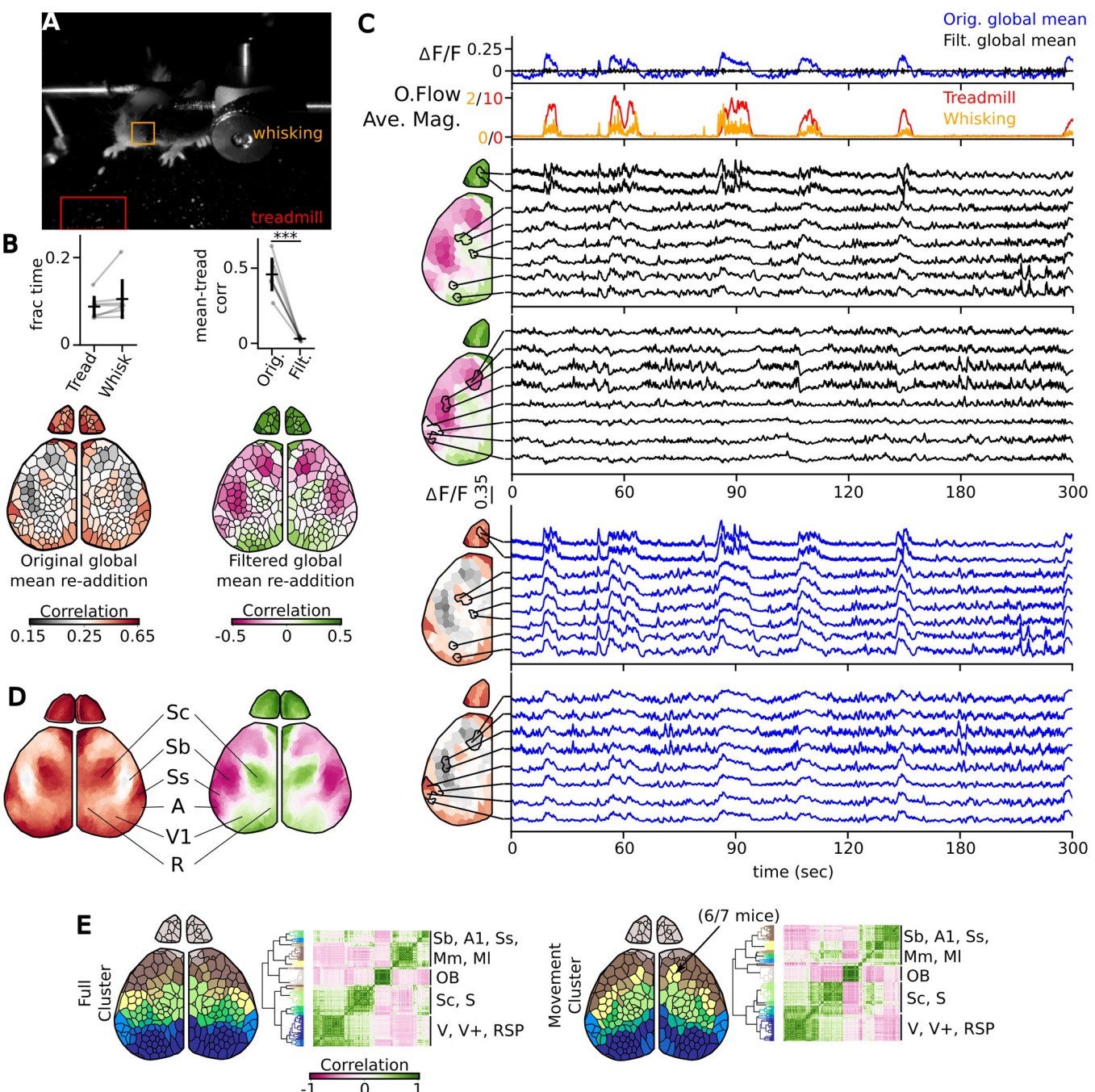

**Fig 10. Validation of domain maps by functional activity during locomotion.** (A) Behavioral video was used to extract locomotion/treadmill (red) and whisking (orange) time course. Boxed regions show analysis via optic flow previously described [41] (B) Correlation domain maps shown with either the re-addition of the original mean (left) and the filtered mean (right). Scatter plot showing the fraction of time walking or whisking (left) and correlation of the global mean to locomotion (right). (C) Comparison of the original mean (top; blue) and filtered mean (black) to the treadmill (red) and whisking (orange) time courses, and individual domain time courses with either the filtered (black) or original (blue) mean re-added that correlated to locomotion or were anticorrelated/less correlated. (D) Average domain map with global mean (left) and filtered mean (right) readded, functionally discrete regions highlighted. (E) Cluster analysis on the full movie (left) or just the times when locomotion was occurring (right). The co-clustering of the frontal motor region with barrel fields occurred in 6/7 mice.

thus the high-pass filtered mean removes more neurologically relevant fluorescence than necessary. However, it has been shown that even if one filters out the global increase, one can still assess similar circuit coherence on wide-field calcium data [40].

With the filtered mean, we noted several regions that increased in correlation and others that decreased (Fig 10C). This result was consistent across all animals tested, as seen in the average domain correlation map (Fig 10D). Domains assigned to regions in primary visual, retrosplenial, and somatosensory core increased in correlation during the motion. Those domains assigned to the somatosensory-barrel fields and auditory cortex decreased. Locomotion or whisking state dependent changes have been observed to decrease the excitability in both auditory [42] and barrel fields [43], while increased coherence has been seen between visual, retrosplenial, and somatosensory core [40]. We were able to validate the assignment of each domain to its respective region, based on which areas of the brain increased and decreased in correlation to the movement vectors.

We performed cluster analysis to determine coherent structures of the full time course and saw nice clustering with bilateral symmetry (Fig 10E). Many of the clusters spanned between the large anterior domains into the smaller domains. We interpret this as the motor/sensory units of the cortex (addressed in the discussion). When we perform the cluster analysis on just the times of locomotion, we were able to cluster the whisking motor unit with the barrel fields in 6/7 of the mice [43,44]. The added benefit of having discrete domains allows researchers to dissect out each functional motor region, which are commonly aggregated together in this type of data collection.

Finally, we wanted to show that no data was lost during any stage of the pipeline and that functional patterns seen corroborate with our assignment to distinct regions (S13 Fig). Point correlation was done on the raw data as well as the filtered data. Artifact sIC were missing structure in the filtered data when the point correlation was plotted against the sIC values. An example of each neural regional sIC is given with its raw and filtered point and domain correlation values. One can compare across the distinct patterns seen and compare results to known activity patterns [45].

## Discussion

Wide-field calcium imaging has grown in popularity in the last decade due to advances in genetically encoded calcium indicators; however, the methods used to isolate neural signal sources have remained underdeveloped [23]. Here we use an ICA-based processing pipeline that removes many of the numerous artifacts that complicate data analysis, including vascular and movement artifacts. We explore quantitative analysis of parameter selection using the resulting eigendecomposition and show a data-driven informed segmentation of cortical patches in an animal dependent manner. These analyses will allow researchers to explore datasets of any age, treatment, genotype, or strain that would be impeded by the use of a reference map.

Calcium data gives insight into underlying cytoarchitecture of coactivating neuronal circuits (i.e. meso-circuits) and thus can be segmented based on their function unit. Our goal in this project was to fit a spatial ICA model to segment and account for all patches of the cortex. In the field, mesoscale imaging has been segmented using point correlation studies to identify functional cortical domains in conjunction with viral tracing experiments [8,15,46]. Spike triggered average investigations have mapped both basal ganglia [47] and thalamic [45] circuits, showing distinct functional cortical patterns when stimulating these major subcortical inputs. During motor learning and locomotion, wide field calcium imaging sessions have been decomposed using Joint Approximation Diagonalization of Eigenmatrices (JADE) spatial ICA to identify relevant cortical nodes [40,48,49], resulting in a sparse representation of the cortex

(16–28 ICs were identified per mouse). Finally, local Non-Negative Matrix Factorization (localNMF) was developed to locate functional domains across the full cortex resulting in over 200 components; however, this method relies on the spatial restrictions of subcortical regions based on a reference map [23]. With the FastICA based pipeline used in the present work, we could fit a high order model that includes 200–300 relevant neuronal components. The total number of components is similar to the number of previously reported localNMF components, however the components segmented with this ICA based pipeline are more discrete and resemble known cytoarchitecture and function without the need of a reference map.

Exploration of recording parameters showed little effect with temporal down sampling (Fig 2). FastICA was able to identify similar numbers of components in temporally down sampled data. There is a drastic shift in autocorrelation in the 2.5 Hz sampled recordings due to the severe under sampling of calcium signal frequencies. This decrease in lag-1 autocorrelation shows that adjacent data points lose relevance to each other; further adding to the idea that the majority of IC signal wavelet power is between 1 and 4 Hz (Fig 5). This argues that 10 Hz should be sufficient sampling for this specific age and GCaMP expression, where the nyquist sampling frequency is above the majority of signal power. Though increasing the duration of exposure would increase the signal to noise ratio of each pixel, this increased signal to noise did not impact segmentation.

Down sampling the spatial information resulted in increased numbers of less informative ICs to describe the data. Spatial down sampling is a common method to decrease computational needs, but our analysis shows some of the shortcomings. Increased resolution and the statistics of large numbers of coactivating pixels help the algorithm to identify the neural and confounding signal sources. Increased recording resolution allows the fastICA to identify diffuse artifact signals, while increasing the number of samples to identify neuronal coactivating pixels, albeit with much more noise. Many of the artifact components are small structures that are diffuse, such as small vascular segments or varying vascular displacement signals across the cortex due to mouse movements. Down sampling will combine the artifact signals with the neural signals, thereby increasing the complexity of the signal and, as such, results in ICA identifying more components. Evidence to support this is shown in the increased autocorrelation due to more temporal structure in the noise components with successive down sampling, ultimately resulting in the failure to identify signals from noise components based on the autocorrelation value (Fig 2). It is possible that binned data can increase the differences between patches of adjacent cortex and lead to a distinct IC. Regardless of the reasoning, the increase in spatial down sampling results in the decrease in both the footprint and the temporal variation. Further, we saw evidence of ICA starting to fail at isolating relevant signals as early as 5x5 binning (35um bin size), resulting in spike-like components which are described as byproducts of ICA when it has insufficient samples [24,50]. However, comparable ICs were identifiable between each of the down sampled resolutions and the number of spike-like ICs never surpassed 7% of the total number of ICs, leading us to believe that some of the sIC may still be relevant. The best fit model with fewest components to describe the cortex remains the model fit with the full resolution data resulting in large, temporally complex ICs.

There has been a concerted amount of effort to minimize the contribution of non-neuronal signals in population dynamics in these wide-field calcium imaging sessions. Reducing hemodynamic influence has been a major focus, and both computational and hardware-based solutions have been suggested [12,51]. One commonly used hardware solution is to interleave excitation lights in order to approximate the amount of hemoglobin influence on the activity patterns [10]. Several studies have utilized data that has been corrected in parallel to those that have not, resulting in evidence that hemodynamics have limited influence on the research outcome but increase the variance in the results [8,13,40,48,52,53]. For those that work in

spontaneous activity, further evidence shows that only weak correlations are seen between hemodynamics and GCaMP activity, and are non-existent in early postnatal mice [10,54]. Even with current hemodynamic solutions, many artifacts persist due to the latency between frame collection; these artifacts are not separable with either PCA or localNMF [23]. Scrubbing vascular and movement artifacts from the calcium imaging data will improve data quality and decrease variance, either through manual sorting or automatic machine learning methods. ICA is able to isolate many of the artifacts that are seen in unanesthetized mouse recordings, allowing researchers to remove signal sources which contaminate the neural signal. These components have spatial and temporal features that are distinct and can be easily identified using our custom GUI or machine learning pipeline (Figs 4–7). We and two other labs have successfully implemented an ICA-based filtration to isolate the neural signal from artifacts in whole cortex [40,48,55,56] and cranial window recordings [41]; however, these applications have not been characterized to the full extent as seen in this manuscript.

When applying this method, careful considerations into potential over interpretations of the results should be weighed. Signals that arise from each patch of cortex are a mixture of repeated signals and this method is unable to discern signals with similar properties. As such, ICA is unable to separate layer specific signals and we most likely have residual hemodynamic influence. While we were able to remove the majority of hemodynamics with our vascular artifacts, we never saw similar sIC in our experimental group as in the neuro-like sIC from our GFP controls. We conclude that the signal is intermixed with each neural component. Further, from our wavelet analysis, we saw the majority of calcium neural signal in the 1–4 Hz range, below the range of heart rate (5–14 Hz) but similar to that of the breathing rate (1.3–3.8 Hz) [57]. It is possible that other biological mixed signals persist in each of the neural components.

Maximal segmentation of the cortex was achieved just before 20 minutes of ongoing resting-state non-task driven "spontaneous" activity, resulting in stable numbers of sIC generated from individual animals (Fig 3). While the time was used as the qualifying metric, we believe that sufficient numbers and types of activations greatly influence the resulting set of sIC. A recent publication has shown that when sparse sIC representation methods are used, there is stability in the sIC across days of recordings [49]. In the approach outlined in this paper, allowing a large number of components (and therefore number of domains) will be determined by distinct states adding to the variation between subsequent recordings. If our pipeline were used on task-dependent data, we expect that will shift the preference of each sIC to that active state. In our analysis, we note that decompositions with a reduced amount of time results in large sIC and only with repeated activations results in more stabilized localized spatial patterns. As discussed previously [14], the dispersion of fluorescence and mixtures of signals in this type of data collection have to be considered during the analysis. With fewer events, the sIC is less certain about the independence of each patch of cortex. With increased samples, FastICA increases confidence in identifying a given area. Each sIC is a repeating mixture of signals that result in a functional mesoscale unit, where the majority of the signal comes from the upper layers of the cortex. The high order ICA model built from the full resolution decomposition best captures the limitations of how precise one can be with this technique. Frequently, the resulting sIC had multiple areas of activation, typically due to bilateral symmetry, with one larger domain and higher relative values. When co-activity occurred within the same hemisphere, higher order areas notably either integrate information (ie. secondary somatosensory cortex) or are a product of divergent circuits (ie. higher order visual areas) [36]. Researchers can perform analysis directly on the resulting sIC to help understand specific units of the cortex or can rebuild the filtered biological data from the selected sIC.

We proposed the creation of domain maps, thereby achieving a functional data-driven map that results from maximally projecting through the sIC. Each domain represents a patch of

cortex with the average reconstruction of functional neurological data. Reconstruction causes each domain to be a mixture of sIC, thus we lose the independence of one domain to the next; however, we retain the variety in shape and size of discrete units across the cortical surface (Figs 8 and 9). Most notably, the highly eccentric domains that surround regional borders assist in identifying functional regions. Further, we noted large domains located in the anterior most regions of the cortex and a higher density of sIC in primary sensory regions that help to identify the distinction between sensory and motor areas. Here we see functional evidence motor units are larger and more extensive than their sensory counterparts in the mouse cortex. Several smaller domains arise from sIC that have multiple functional areas and the reconstructed time course is more prone to loose correlation with their sIC temporal feature (Fig 7).

Reconstruction of the biological signal opens up investigations into how much each sIC contributes to each patch of cortex and allows researchers to build more computationally demanding spatial models and perform time series analysis that would otherwise be impossible based on a pixel-wise analysis [35,58–60]. We were able to perform a wavelet coherence and event-based analysis utilizing these maps, allowing us to help elucidate changes across different experimental groups in cortical functional and spatial structure [56].

The full reconstruction of the neurological data requires the re-addition of the global mean, but removing artifact components complicates its interpretation. As shown, the global median is highly correlated both to locomotion and preferentially to many artifact sIC (Figs 10 and S8). Since the artifacts influenced the global mean, removal of all artifacts requires the modification of the mean before re-addition. In this manuscript, we propose the high-pass filtration of the slow frequencies (0.5Hz cutoff; S9 Fig). Many studies have observed an increase in cortical activity based on locomotion [39,40]; therefore, in addition to the artifact influence we are also removing neurologically relevant fluorescence. With the filtered mean re-added, we observe a wide collection of correlation values, with the original global mean re-added, the possible values are (S13 Fig). Results from locomotion coherence studies have shown similar results from both filtered and unfiltered comparisons [40], and as we show there are similarities between the coherence analysis done on both the original global mean and the filtered mean domain clustering (Fig 10).

Functional imaging in unanesthetized, behaving animals gives insight into the nature of physiological processes; however, nontrivial challenges arise during such sessions with intermixed sets of time varying signals. The methods presented here address the most common issues in analyzing large wide-field mesoscale datasets, including filtration of vessel artifacts, spatial mapping, and optimized time series analysis. This work demonstrates that signal components, having maximal statistical independence captured in sufficiently sampled monochromatic calcium flux videos, exhibit a combination of spatiotemporal features that allow machine classification of signal type. Implementation of automated machine classifiers for neural signals is practical given densely captured arrays of spatially and temporally variant data gathered from individual subjects. With these tools, neuroscientists can easily collect and analyze high quality neural dynamics across the cortical surface, allowing the investigation of complex networks at unprecedented scale.

## Materials and methods

### Ethics statement

All animal studies were conducted in accordance with the University of California, Santa Cruz Office of Animal Research Oversight and Institutional Animal Care and Use Committee under protocol numbers Ackmj1507 and Ackmj1807.

## Mice

P21-22 Snap25 GCaMP6s transgenic mice (JAX: 025111), Cx3cr1 GFP (JAX: 005582), and Aldh1 GFP (MGI: 3843271) were maintained on a C57/Bl6 background in UCSCs mouse facilities. To identify Snap25 GCaMP expressing mice, a single common forward primer (5'-CCC AGT TGA GAT TGG AAA GTG-3') was used in conjunction with either transgene specific reverse primer (5'-ACT TCG CAC AGG ATC CAA GA-3'; 230 band size) or control reverse primer (5'-CTG GTT TTG TTG GAA TCA GC-3'; 498 band size). The expression of this transgene resulted in pan-neuronal expression of GCaMP6s throughout the nervous system. To identify GFP expressing mice a forward (5'-CCT ACG GCG TGC AGT GCT TCA GC-3') and reverse (5'-CGG CGA GCT GCA CGC TGC GTC CTC-3'; 400 band size) PCR amplification was used to identify which animals had the GFP transgene. At the end of each recording session, the animal was either euthanized or perfused and the brain dissected.

7 animals used in this study were to experiment and control mice to study perinatal penicillin exposure effects on cerebral networks [56]. These methods work independent of experimental conditions and the perinatal penicillin had little effect on domain parcellation.

## Surgical procedure

All mice were anesthetized with isoflurane (2.5% in pure oxygen) for the procedure. Body temperature was maintained at 30C for the duration of the surgery and recovery using a feedback-regulated heating pad. Lidocaine (1%) was applied subcutaneously in the scalp, followed by careful removal of skin above the skull. Ophthalmic ointment was used to protect the eyes during the surgery. The cranium was attached to two head bars using cyanoacrylate, one across the occipital bone of the skull and the other on the lateral parietal bone. After the surgery was complete, mice were transferred to a rotating disk for the duration of the recording which started 1 hour after the last isoflurane was administered. At the end of the recording session, the animal was euthanized and the brain dissected.

## Recording calcium dynamics

In-vivo wide-field fluorescence recordings were collected in a minimally invasive manner. Imaging through the skull by single-photon excitation light from two blue LED light (470 nm; Thorlabs M470L3) produces a green fluorescent signal that is collected through coupled 50mm Nikon lenses (f5.6 / f1.2, optical magnification $\sim$ 1x) into a scientific CMOS camera (PCO Edge 5.5MP, 6.5μm pixel resolution). The top lens in the tandem lens system was used to focus on the cortical surface, thereby lowering the magnification slightly; anatomical representation for each pixel corresponded to 6.9±0.2μm (min: 6.7μm, max: 7.2μm). Excitation light was filtered with a 480/30 nm bandpass (Chroma Technology AT480/30x) and the emission signal was filtered with 520/36 nm bandpass (Edmund Optics 67–044). Data collection was performed in a dark, quiet room with minimal changes in ambient light or sound, thus the brain activity recorded was at resting state without direct stimulation. Raw data was written directly as a set of 16 bit multi-image TIFF files. Concurrent body camera recording was taken using infrared (IR) raspberry pi cameras (Raspberry Pi NoIR V2), recorded at 30 fps synced to the CMOS.

The total amount of data recorded for each animal was generally at least 40 min and the amount of time in between video segments was less than 1 minute. All analyzed data consisted of two sequential full spatial resolution recordings concatenated together giving 20 min of data sampled at 10 frames per second for each video decomposition.

All video segments consisted of a set of continuously collected images at 10 or 20 frames per second for 10 minutes. The total amount of recorded data for each animal was generally at

least 40 min and the amount of time in between video segments was less than 1 minute. When more than 10 minutes of video data was used for single decompositions, multiple videos were concatenated together. All analyses were conducted on data recorded at 10Hz, except exploration of effects of resolution on data quality.

Spatial resolution analyses were performed on a single 10 minute recording at 10Hz. Spatial down sampling was conducted by taking the mean between groups of pixels in a dxd square, where d is the integer down sampling factor. Temporal resolution analyses were performed on a single 10 minute recording at 20Hz. This data was temporally down sampled by taking the mean between an integer number of subsequent frames.

## ICA decomposition and saving

ICA was performed using FastICA [24], implemented through python's sklearn decomposition [61]. The ICA decomposition was applied to the spatially flattened (xy,t) 2-D representation of the video data under the cortical ROI mask. The mean time series is pre-subtracted from the array before SVD decomposition or ICA decomposition, since ICA cannot separate sources with a mean signal effect. The filtered, unfiltered mean, ICA components, mixing matrix, and associated metadata are all saved. Data is stored and saved in this flattened format for storage optimization. Components are locally spatially reconstructed for visualization in the GUI.

Requesting the full number of components resulted in extremely lengthy processing times. To reduce the processing time, the data was preprocessed through Singular Value Decomposition (SVD) whitening, and noise components were cropped. To ensure that no signal was lost, and there were ample dimensions left for ICA separation, the inflection point between SVD signal and noise floor was identified, and SVD components were reduced to include components equal to 5 times the SVD signal to noise cutoff value. This cutoff multiplier can be adjusted while ICA projecting.

After calculating and sorting the ICA results, excessive noise components are removed from the dataset for compression. The cutoff was determined by identifying the inflection point in the lag-1 autocorrelation distribution with a two-peaked KDE fit. Components were saved such that 75% of the components saved were signal or artifact, and 25% of the components saved were noise associated. If not enough noise components were returned by the ICA decomposition, there is a risk that signals were not sufficiently unmixed, so the ICA decomposition was repeated with a higher SVD cutoff until enough additional noise components were included. This was not necessary for any of our experiments in this paper, but can be useful in ensuring signal capture in recordings of smaller areas, such as cranial windows.

For resolution down sampling analysis, the number of components requested was kept constant for all resolution values to avoid confounding the results. The cutoff for the highest resolution tested was used for all lower resolution decompositions.

ICA returns components that are unsorted and often spatiotemporally inverted. Components were first sorted by their time series variance. The spatial histogram of sIC values across each component can be visualized as a single tailed gaussian distribution centered around 0, where the tail represents the spatial domain of each component. The two edges of the distribution are first identified. The boundary closer to 0 is taken as the edge of the central noise distribution, and that boundary is used to define the dynamic threshold on the opposite side of 0. Any values outside of this noise distribution and is part of the wider tail are included in the binarized domain of the component. If the tail was negative, the component was inverted spatially and temporally by multiplying each by a factor of -1. In this way, components were all

identified as positive effectors for visualization purposes. Movie rebuilding is not affected by this process.

All code used in this paper is available at github.com/ackmanlab/pyseas or as a package (pySEAS: python Signal Extraction and Segmentation) in the python package index (pip install seas).

### Dynamic thresholding

The spatial histogram of sIC values across each component can be visualized as a single tailed gaussian distribution centered around 0, where the tail represents the spatial footprint of each component. The two edges of the distribution are first identified. The boundary closer to 0 is taken as the edge of the central noise distribution, and that boundary is used to define the dynamic threshold on the opposite side of 0. Any values outside of this noise distribution and is part of the wider tail are included in the binarized domain of the component. If the tail was negative, the component was flipped spatially and temporally for visualization purposes.

### Data processing

ICA decompositions of videos at full spatial resolution and duration (20 min) were run on a single CPU node of a computing cluster having 1024 GB of RAM. Shorter recording length videos (10 min) could be processed on a node having 512 GB of RAM. After ICA processing, map creation and time series analysis were performed on local computers having 16-32GB of RAM.

### Metric generation and classification of Neural Independent Components

An ensemble random forest classifier from the scikit-learn packages [62,63] was used to train and classify between human scored signal and artifact components [64], based on features calculated from each component. Wavelet Mean filtration Wavelet decomposition on the time series signals were performed with a ω = 4 Morlet wavelet family, code adapted from C. Torrence and G. Compo [65] available at URL: http://paos.colorado.edu/research/wavelets/ Significance was determined using the 95th percentile of a red-noise model fit to the time series autocorrelation. Frequency distributions are all displayed as the ratio of the global wavelet spectrum, relative to the noise cutoff. For wavelet filtering, the original signal was rebuilt excluding all frequency signals in a certain range.

### Map creation and comparisons

Domain maps were created by separating the cortex into regions represented by different ICA components. Each component was blurred by a 51-pixel kernel, then the maximum projection was taken through the component layers. The resulting data is a cortical map that denotes the component with maximum influence over any given pixel.

This map was then further processed to get rid of domains smaller than 1/10th the mean domain. Any domain smaller than this size is checked to see if the second most significant component would produce a larger continuous structure. If after a few loops of this, pixels cannot be assigned into a larger structure, the points are excluded from the final map. Indices are then adjusted such that any non-continuous regions represented by the same domain are assigned to different units.

Olfactory bulbs were included in map generation, but domains were highly variable, and were excluded from map quantification metrics.

Voronoi maps were created to match the same number of domains or regions as the original map, n, and shares the same cortical mask as the original map. To create this map, n points were distributed randomly across the cortical mask. To turn these points into regions, the voronoi diagram was created using the scipy spatial package and was applied as a voronoi map.

Grid maps were created to match or exceed the same number of units as the original map, and share the same cortical mask as the original map. A uniformly spaced 2D grid was placed over the original map, and resulting units were counted. If the number of resulting spatial units exceeded that of the original map by $< 15$, the map was accepted as a valid comparison. Otherwise, the map was rejected and a new grid map was calculated.

For every domain or region in the original map, the nearest neighbor was identified in the comparison map with a KNN tree. To quantify the spatial similarity of each identified domain or region, the Jaccard index (spatial overlap / union) was then calculated. For each comparison, n Jaccard indices were calculated.

When comparing maps generated from different animals, the optimal alignment was calculated by shifting the second map up to 100 pixels in any direction. The optimal direction was determined by maximizing the Jaccard overlap. Each generated map was compared to one map from the same animal, one littermate, and two non-littermates, as well as one randomly generated voronoi map.

## Component correlation to movement

To define the movement vector, we used an OpenCV grid-based optic flow algorithm to calculate the motion magnitude on the body camera data. Down sampling of the time course (from 30 to 10 fps) was done and the Pearson coefficient was produced with respect to each components' time series.

## Compression and filtering residuals

Compression residuals are calculated while saving the ICA decomposition results. The original movie is rebuilt from the reduced ICA results, and residuals are calculated by taking the absolute value of the difference between the two videos (S14 Fig). The spatial and temporal projection of this absolute difference movie is saved as the spatial and temporal residuals of the decomposition, and is stored as metadata with each ICA decomposition.

## Domain residuals and domain signal analyses

To quantify the amount of signal present in the original movie that was not included in the domain map, residuals were calculated by subtracting the 'mosaic movie', representing time series from each spatial domain from the original movie. The absolute value was then applied so that all numbers represented a positive difference, and residuals were summed to create a single value. The time series was not re-added to either the original movie or mosaic movie, since this can be easily summarized as a different temporal metric. To represent the amount of relative variation to the original dataset, this number was compared to the summed absolute value of the mean-subtracted original movie.

## Statistical significance

Statistical significance was calculated using OLS models from statsmodel.formula.api with Holm-Sidak multiple testing correction ($p \leq 0.5$: *; $p \leq 0.01$: **; $p \leq 0.001$: ***). Model significance is determined by the F-statistic, and significance of two-group analyses ($p > |t|$) are calculated with t-tests.

## Dataset availability

Relevant datasets have been uploaded to a Dryad repository [66].

## Dryad DOI

https://doi.org/10.7291/D1N96W

## Supporting information

**S1 Fig. A Tkinter-based graphical user interface (GUI) for browsing independent component analysis results.** (A) 15 independent components, order 60–74 by variance. Components displayed in gray are manually selected as artifact either manually or using a machine learning classifier. A click on the display for any given component manually toggles its classification as either signal or artifact associated. Components colored in the cool/warm colormap are neural associated. Components colored in the black/white colormap are artifact associated. Buttons on the bottom panel control GUI movement through the dataset. The text panel at the bottom displays the index used as the signal/noise cutoff. (B) The component viewer displays additional temporal metrics for any given component. The top controls allow movement through the dataset by manual scrolling with (+/-) buttons, up/down keys, or through typing a desired component in the text box. sIC time course displays the mixing matrix time course extracted by ICA for the given components. The Wavelet power spectrum is displayed in the bottom right, and an integrated wavelet or Fourier representation is available on the bottom left. 0.95 significance as estimated by the AR(1) autoregressive red-noise null hypothesis is displayed as a dot-dash line. (C) The domain map correlation page shows the Pearson's correlation coefficient between a selected seed domain and every other domain detected on the cortical surface. The seed domain can be changed through the arrow keys, the (+/-) buttons, or by clicking on a different domain on the displayed domain map. (D) The Component region assignment page allows manual region assignment for each domain. After the region is selected from the menu on the right, each domain clicked on the domain map is assigned to that region.
(TIF)

**S2 Fig. Comparable IC examples from spatial down sampling experiment.** (A) Comparable examples of neuronal (top) and artifact (bottom) sIC (left) with their corresponding time series plotted on top of each other (right). Moving left to right shows a decrease in the spatial sampling rate of the decomposition. (B) Emergence of spike-like sICs were seen after 35um down sampling.
(TIF)

**S3 Fig. Comparable IC examples from temporal down sampling experiment.** Comparable examples of neuronal (top) and artifact (bottom) sIC (left) with their corresponding time series plotted on top of each other (right). Moving left to right shows a decrease in the temporal sampling rate of the decomposition.
(TIF)

**S4 Fig. Wavelet transform can be used to generate a signal-to-noise ratio that indicates significant frequencies.** (A) Example neural time series, 90 sec of data recorded at 10hz reported in the temporal portion of a component (B) Morlet wavelet ($\omega = 4$) was used for the wavelet transform. (C) The power spectra of the wavelet transform (colorbar, blue to yellow) and the global spectral analysis (black, right). The 95% quantile is shown in dashed lines on the global spectral analysis. Reformatting the frequency spacing, produces. (D) A power signal-to-noise ratio is calculated by dividing the power spectra by the 95% quantile of red noise defined by

the AR(1). All values above 1 (dashed line) would indicate a high probability of signal. Anything below 1 would most likely be considered noise (colorbar, green to red). (F) Reformatted frequency spacing.
(TIF)

**S5 Fig. Spatial, Morphometric, Temporal, and Frequency features extracted from components.** (A) Spatial metrics from statistical characteristics of each sIC (spatial representation of the component). The histogram of all sIC values is shown the right of the sIC. (B) Morphometrics collected from the binarized thresholded masked region of the sIC. The largest (primary) domain was used to generate the features for each sIC. The majority of metrics calculated utilizes sci-kit image region properties. (C) Temporal metrics are statistical descriptors from the corresponding row of the mixing matrix for each sIC. (D) Frequency analysis was done on the mixing matrix row, utilizing the PNR calculated from wavelet analysis (S4 Fig). The longest of all continuous frequencies was used to extract each feature.
(TIF)

**S6 Fig. Examples of control components, resulting in similar artifact components to GCaMP recordings.** (A) Control components from 20 minutes of recording from cx3cr1 GFP (microglia; mGFP, left), adlh1 GFP (astrocyte; aGFP, center), and Black 6 (Non-transgenic; Bl6, right) mice. Two IC examples from each control group corresponding to hemodynamics/ neural activity (top) and artifacts (bottom). Artifacts chosen show a vascular and other artifact commonly seen in GCaMP recordings. Similar data description in regards to temporal and spatial representations as seen in Fig 1. (B) Examples of control binarization of the sIC only showing the windowed spatial representation on the key portions of the sIC. (C) Examples of neural and artifact wavelet analysis shown in the power signal-to-noise ratio (PNR) plots.
(TIF)

**S7 Fig. Great machine learning performance with multiple classification algorithms.** A) Receiver operating characteristic (ROC) plots for each classifying algorithm utilized. Voting classifier was composed of the 4 other algorithms. Random Forest Classifier (RFC; *) was used in all analyses in this paper. B) Histogram of human classification with the percent and error of each training iteration, binned based on confidence (same data as Fig 6F, bottom). Log-scale was used to highlight the low percentage points. C) Histogram of human classification with the percent in each binned novel classification. Classification bins based on the percent each classification occurred correctly in the 1000 trials. True positive (TP), False positive (FP), False negative (FN), True Negative (TN), human (h), machine (m).
(TIF)

**S8 Fig. Mean filtration to minimize global slow oscillations seen in GFP control data.** (A) 30sec examples of the global mean that was subtracted and stored at the initiation of the pipeline, prior to before the decomposition into sIC for GCaMP, mGFP, aGFP and Bl6. (B) Global wavelet spectrum (top) and its corresponding power to noise ratio (PNR; bottom) of GCaMP (N = 4), mGFP (N = 3), aGFP (N = 3), and Bl6 (N = 3), red indicates the omitted frequencies from our applied high pass filter. (C) High-pass filtration results of the same 30 sec in A.
(TIF)

**S9 Fig. Vascular and other artifacts are more correlated to movement than neural components.** (A) All neural (blue), vascular (red), and other (orange) components and their correlation to the motion vector from each animal. (B) Spatial location and corresponding correlation (green to pink) of each component to motion based on their respective classification and genetic background. neural: left column, vascular: center column, other: right

column. Top row: GCaMP, second row: mGFP, third row: aGFP, last row: Bl6.
(TIF)

**S10 Fig. Point correlation analysis examples.** Corresponding spatial ICs matched to the time-series domains shown in Fig 7(F). Point correlation based on the seed location of the maximal value of each IC shown in pink and green. Difference of the point correlation (gray and red) from all adjacent ICs (bottom 7E) with respect to the center domain (top domain). Values from each point correlation and difference were used in Fig 7E.
(TIF)

**S11 Fig. Individual domain maps from 20min, 10 Hz, full resolution animals used in this study.** (A) Domain maps generated from Snap25 GCaMP6s recordings from littermates (a-c) and from subsequent recordings (#). (B) Domain maps generated from the three different control lines.
(TIF)

**S12 Fig. Individual domain maps from down sampled and duration experiments.** (A) Domain maps made from spatial and (B) temporal down sampling experiments. (C) Domain maps from duration experiments at 5 duration time points.
(TIF)

**S13 Fig. IC, Point correlation, Filtration, Domain map comparison across regions.** Comparison of sIC, raw data and filtered pixel wise point correlation across 2 examples of artifacts and 11 neuronal sIC, each from a distinct region of the brain. Each point correlation map was created based on the location of the maxima of the sIC (indicated by the circled regions of the point correlation maps), either using the raw data (gray to red) or the filtered data (pink to green). Pixel-wise comparison of how each correlation map plots to the IC value plotted to the right of each correlation map. Open triangles (top 2 rows, right) denote the absence of structure in the scatter plot of the artifact ICs compared to the filtered correlation map. Domain correlation maps were created by mapping each IC to which domain it most contributed, and used that domain as the seed correlation. Domain maps with gray to red colormaps have re-addition of the original mean and pink to green colormaps have re-addition of the filtered mean.
(TIF)

**S14 Fig. Comparison of spatial and temporal information content through compression and filtering.** (A) The original spatial information captured as quantified by a mean subtracted absolute value projected spatially (left) or temporally (right). (B) The difference in information between the original input data and the rebuilt ICA projection, excluding noise components beyond the 25% saved in the processed file. The difference movie is projected spatially or temporally to visualize where information was lost in compression. (C) Information removed by artifact filter. The artifact movie is rebuilt and projected spatially or temporally to visualize where information was modified by the ICA-based artifact filter.
(TIF)

**S1 Video. ICA filtration removes artifacts for superior neural signal unmixing.** Original video (left) is decomposed into neural and artifact components. The filtered artifact movie (center) can be rebuilt to visualize artifacts that were isolated and removed. The rebuilt neural signal (right) depicts just the filtered neural signal. A high pass filtered mean (0.5Hz cutoff) is re-added to both filtered artifact and neural signals. Video is a real-time 1-minute excerpt. Values displayed are in ΔF/F.
(MP4)

**S2 Video. Re-addition of the global mean with high-pass filter represents neural signal.** Rebuilt video from all components (left) or with only neural components (center, right) with mean re-addition from the original global mean (left, center) or the global mean with 0.5 Hz high pass filtration (right). All movies are on the same scale of change in fluorescence over mean fluorescence (rainbow colorbar).
(MP4)

**S3 Video. Mosaic movie represents the neural signal captured from domain time series across the cortical surface.** Neural signal from the filtered video (left) is segmented by the data-driven domain map (left overlay). Average time series from these segmented domains can be visualized as a mosaic movie, where each domain is represented by its averaged time series. The filtered video contains 1.77 megapixels representing the cortical signal, while the mosaic movie contains only 300 unique time series to describe the same signal with a 5900x compression rate. Video is a real-time 1-minute excerpt. Values displayed are in ΔF/F.
(MP4)

## Acknowledgments

The authors acknowledge C. Santo Thomas for maintaining the lab mouse lines, and University of California Santa Cruz's Hummingbird Computational Cluster for support and node maintenance. We thank David Feldheim, Jena Yamada, and Dan Evans-Turner for proof-reading the manuscript.

## Contributions

ICA filtering, exploratory GUI, map creation and time series extraction and analysis code, was written by S.C.W. All recordings, metric extractions, mean frequency analysis, feature extraction and analysis, machine learning pipeline were performed by B.R.M. Optimizing and determination of hyperparameters was done by D.A. J.B.A. oversaw the project and provided feedback to experimental design, results, and paper preparation. The manuscript was prepared by B.R.M. and S.C.W, with input from all authors.

## Author Contributions

**Conceptualization:** James B. Ackman.

**Data curation:** Brian R. Mullen.

**Formal analysis:** Sydney C. Weiser, Brian R. Mullen.

**Funding acquisition:** James B. Ackman.

**Investigation:** Brian R. Mullen, Desiderio Ascencio.

**Methodology:** Sydney C. Weiser, Brian R. Mullen.

**Project administration:** James B. Ackman.

**Resources:** James B. Ackman.

**Software:** Sydney C. Weiser, Brian R. Mullen.

**Supervision:** James B. Ackman.

**Validation:** Desiderio Ascencio.

**Visualization:** Sydney C. Weiser, Brian R. Mullen.

**Writing – original draft:** Sydney C. Weiser, Brian R. Mullen, James B. Ackman.

**Writing – review & editing:** Sydney C. Weiser, Brian R. Mullen, James B. Ackman.

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
