## [Decision Letter · Decision Letter 0]

27 Oct 2022

Dear Dr Mullen,

Thank you very much for submitting your manuscript "Data-driven segmentation of cortical calcium dynamics" for consideration at PLOS Computational Biology.

As with all papers reviewed by the journal, your manuscript was reviewed by members of the editorial board and by several independent reviewers. In light of the reviews (below this email), we would like to invite the resubmission of a significantly-revised version that takes into account the reviewers' comments.

Dear Dr. Mullen,

As you will see both reviewers are raising significant issues that you will need to address before a final decision can be made. As you will read, the significance of your contribution would be much greater if the functional maps that you are extracting could be confirmed. There is clearly a lingering doubt in the mind of both reviewers that the actual details in your methodology would shape these results.

I look forward to your reply.

Frederic Theunissen

We cannot make any decision about publication until we have seen the revised manuscript and your response to the reviewers' comments. Your revised manuscript is also likely to be sent to reviewers for further evaluation.

Sincerely,

Frédéric E. Theunissen

Academic Editor

PLOS Computational Biology

Lyle J. Graham

Section Editor

PLOS Computational Biology

Dear Dr. Mullen,

As you will see both reviewers are raising significant issues that you will need to address before a final decision can be made. As you will read, the significance of your contribution would be much greater if the functional maps that you are extracting could be confirmed. There is clearly a lingering doubt in the mind of both reviewers that the actual details in your methodology would shape these results.

I look forward to your reply.

Frederic Theunissen

Reviewer's Responses to Questions

**Comments to the Authors:**

Reviewer #1: Calcium imaging approaches allow to investigate the neuronal activity at a wide range of spatial resolutions, from individual synapses to entire brains. Mesoscale imaging comprises the acquisition of a large portion of the brain. It is widely used in mice that express a genetic calcium indicator in neurons or projections located in superficial regions (cerebral cortex, superior and inferior colliculus). This approach allows to study, for instance, the evoked and spontaneous pattern of activity in the different cortical areas and their interactions. Because experiments are done in vivo, usually without anesthesia, the signal obtained from the movies is far from clean. Many labs obtain data using this imaging approach and usually they develop as well their own pipelines of analysis to process the images, extract data and raise conclusions.

In this manuscript, Wieser, Mullen and colleagues provide an interesting and polished workflow for analyzing mesoscale calcium imaging data in adult mice. The method applies different strategies to increase the quality of neuronal signals and remove artifacts and noise. The method is based on a combination of principal and independent component analysis (PCA and ICA). The components are classified as neuronal, artifact or noise signal using a machine learning approach. In the manuscript, there is also an interesting exploration of the relationship between the performance of the ICA and the duration, spatial and temporal resolution of movies. In addition, the method provides a good way for filtering, compressing as well as segmenting the data. Using this segmentation and additional information, the authors construct animal-specific functional maps that are robust throughout time and litter.

Calcium imaging analysis pipelines based on matrix factorization techniques, and based on ICA specifically, are not new in the field, as the authors state in the Discussion (lines 692 to 691). Nevertheless, one relevant point here is the code that accompanies the manuscript. The authors make a valuable contribution to the community making accessible a python toolbox, PySEAS, for the implementation of their method. The toolbox includes a well-design and friendly GUI.

The method seems to properly identify different types of signals allowing the isolation of cleaner neuronal components. This is the main strength of the manuscript; however, I do have important questions and comments on the procedure and, also, on some statements raised by the authors. In addition, I have some major comments on how the authors build the functional maps of the cortical territory. Revealing functional maps is presented here as one of the actual applications of the method, therefore, the authors must provide arguments as convincing as possible.

Major points

1. One of the main advantages of ICA is the dimensional reduction and filtration possibilities offered by this method, as the authors comment in the manuscript. Using ICA for making a low-dimensional representation of calcium imaging data in which artifacts have been excluded simplifies any subsequent analysis. In this case, the authors show one possible application by using the components to create functional maps of the cortex. I’m not convinced about how these maps are built but, first, I would like to raise another point. I think that the authors could try to link the independent components to physiological or biophysical concepts. For example, the independent components obtained from LFP recordings, also considered as generators of activity, have been associated to specific synaptic pathways or to different locations of the signal source (Makarov et al 2010, Herreras et al 2015). Moreover, independent components from fMRI data are related to functional networks. In other words, it would be interesting to have the opinion of the authors about why ICA is an a-priori suitable tool to analyze this type of data. In this line, it might be interesting to add some lines that justify the main assumption of statistical independence impose on the spatial dimension of the data.

2. The method presented in this manuscript nicely segregates calcium imaging signals increasing the quality of the neuronal component. In my opinion, this is the main achievement of the authors. To illustrate potential applications, the authors describe how the information and segmentation obtained after processing the images could be used to parcellate the cortex and build functional maps. This application would be extremely valuable for neuroscientists if the parcellation is accomplished based on the domain features or any other data-driven information. But, here, the authors manually divide the cortex in regions not only using domain features but also a reference map based on the Allen Brain atlas. Including the reference map decimates the relevance of the method for parcellating the cortex in functional regions. Actually, neuroscientists need a method to build functional maps without a reference map. As such, maps derived from control and non-canonical scenarios (development, manipulations, plasticity, pathologies, etc) could be directly and objectively compared. At the end, the reference maps should be used to validate the analysis workflow since reference maps must resemble the data-driven functional maps of the cortex derived from control mice. Is it possible to create, manually or automatically, functional maps based only in “network analysis” and “metric comparison” (line 621), or other information that does not depend on a standard reference map? If this is the case, data-driven maps should be validated analyzing to what extent they resemble already known maps. I think this is the way to reveal a convincing and relevant application of the method.

3. Spatial independent components are static modes and may not adequately represent the joint spatiotemporal nature of the data. In this kind of experiments, it is common to observe transient events spreading in time and space, with distinct initiation sites and wave-like flows across the cortex. I think that the static nature of spatial independent components and domain maps does not contemplate this information … is it possible to recover such information from the independent components? If not, ICA-based methods could result insufficient for those experimental questions focused on the individual properties of calcium events, especially those related with their frequency and dynamics. Could the authors discuss these observations?

4. Continuing with the quantification of the impact of spatial, temporal resolution and duration on ICA performance, although I appreciate the attempt for creating guiding principles for the acquisition of calcium imaging data that can be adequately filtered and disentangled using ICA, I think that there are some concepts that are not clear enough in the explanation provided by the authors. They conclude that the spatial resolution of the images is a key factor for obtaining a clear-cut separation of the different types of signals. However, as far as I know, I would say that the key factor is the number of pixels (the amount of information) and not the distance between two pixels (spatial resolution), two aspects that in their experimental design are related but not identical. In the down-sampling process for decreasing the spatial resolution the number of pixels is also reduced (even faster than the size of the pixel). By reducing the number of pixels while keeping constant the number of components, at certain point, the sample size will not be enough to properly estimate a given number of independent components (Särelä and Vigário 2003). For instance, to test if this is the case, the authors could check if spike-like components appear when the spatial resolution is low. Spike-like components are artefacts generated by ICA with insufficient sample size (Hyvärinen et al 1999). Also, down-sampling using a binning method not only changes the number of pixels but also the signal to noise ratio. If this is the case, are the results obtained at different binning factors comparable to raise conclusions only about the relationship between ICA performance and number of pixels?

5. Related to the previous point, do the authors think that the performance of ICA to segregate signals depends on the signal-to-noise ratio or the bit range of the signal? Is it possible to explore these dependencies?

6. Regarding the temporal resolution of the data that feed ICA, the authors conclude that it doesn't impact on the segregation of the types of signals as dramatically as spatial resolution. I would say that this is expected since the authors are running a spatial ICA. In the case of spatial ICA, each frame or image of the data is a mixture of source images that are statistically independent. In the spatial ICA model, the constraint of statistical independence is on the sources (the image vectors), while the mixing matrix or temporal courses are unconstrained. Probably, it is not relevant whether temporal courses are shorter (low temporal resolution) or longer (high temporal resolution), affecting only the number of components that can be extracted. In line with this, why does the distribution of lag-1 autocorrelations shift when time resolution is low?

7. In the Discussion, in general, the authors mainly recapitulate the Results and briefly link them to other publications. Some paragraphs, as the one starting in line 708, are just a summary of the Results. I think that more elaboration is needed to explain how the data presented here enters in the conversation with the published literature of the field. Just one example, in the Author Summary, they mention that the signal from neurons from different depths is a confounding factor in this type of experiments. I think that the method presented in this manuscript does not get rid of this problem. However, the authors do not discuss further this particular issue, and general limitations of the technique are vaguely mentioned in lines 705-706. Please, elaborate on knowledge/technical gaps that have been filled, applications, limitations of the method, etc.

Minor comments

1. I’m not sure if the term eigenvector is correct in this case. I’m not sure if independent components are proper eigenvectors, as it is the case for principal components analysis where compinents are the eigenvectors of the cov-matrix of the data. Is this correct?

2. In line 182, the authors mention PC (Principal Component) when in fact they mean “independent component”. And I think that this mistake appears also in Figure Supplementary 1, where they label the different panels as Principal Component Analysis: “PCA viewer”, “Select principal component analysis”. Please, clarify these issues.

3. The use of terms “noise” and “artifact” is sometimes confusing, as in lines 256-261 and related figures. Please, revise the whole text on this regard.

4. In line 100, bibliographic reference is incorrect, 23 instead of 22.

5. In Figure 1, caption mentions a dashed line in panel A that is not present or visible.

6. In line 319, the omega symbol is missing.

7. In line 78-79, “NNNxNNN” pixels is not the spatial resolution.

8. In Figure 1C and others, the scale of Relative Intensity is not the same in the brain schemes and in the traces.

9. In line 205, please rephrase “High resolution spontaneous activity”.

10. In Figure 2D, mean artefact curve does not seem to level off.

11. In line 320, replace purple by blue and panel letters seem to be missing or wrong in Figure Supplementary 2.

12. Please, revise the references in the text for Figure 5F, Supplementary 5A and C, and Supplementary 6C.

13. I think that the caption of Figures Supplementary 6 and 7 are swapped.

14. In general, in the figures showing neural independent components (Figures 1C, 3A, and Supplementary 1) there are negativities (in blue) surrounding the positive part of the component (in red). Which is the origin and interpretation of these negative values?

15. The sixth paragraph of the discussion is hard to read and understand.

Reviewer #2: The authors provide potentially novel work using a higher resolution form of typical mesoscale wide field imaging coupled with analysis pipelines to recover specific networks. The work includes using 7 micrometer pixels but still employing a relatively standard low resolution mesoscale wide field imaging setup. The images undergo extensive processing using an independent component analysis (ICA) pipeline to break up time series images into components elements. The ICA procedure is done for a snap 25 gcamp mouse as well as several different control line mice that express gfp in astrocytes or microglia. The control lines expressing gfp only are put forward as a means of parsing out signal versus artifacts due to pixel noise, vessels or brain movement. Overall the findings are surprising in that the widefield images can be decomposed into many more sub-image domains that were then were previously reported with other methods including seq nmf (Buschman lab). The sub regions are then clustered into large groupings that tend to be consistent with large scale anatomical relations.

The work would be signfiicant and novel if the small sub-regions that are identified can be verified to overlap with known functional activations and confirmed in some way to be units on their own rather than some by-product of the analysis.

Points

Previous work on the resolution of wide field imaging suggests resolution that is likely insufficient to resolve the small clusters that are observed. https://elifesciences.org/articles/59841 see Fig 2 how do the small activations compare with this expected resolution?

Figure 1 examines potential motifs extracted using the ICA procedure. Surprisingly, high resolution (small ½ width) and small areas thought to be neuronal components are shown in panel c and other components shown to be artifacts are also indicated. It would be important to show that these small discrete areas thought to reflect real signals are both within the resolution of wide field imaging and that these areas also represent independently functioning cortical areas if time plots from different roi's are shown. They could also do something like a seed pixel local correlation analysis again to provide information that these areas are indeed independent elements as they proposed. Thus far the evidence that these are unique regions with independent function is not convincing.

Figure 2 shows the variance of the eigenvector's and is used to argue that neural signals are indeed separated from artifacts based on their autocorrelation and their variance profiles. However, to do this one needs to assume that artifacts and neural activity operate on significantly different time scales and spatial properties. In the case of vascular noise this could potentially be on a slow second time scale similar to neuronal activity or breathing or large body movements.

Figure 3. Demonstrates the recovery of potential activity like motifs within gcamp expressing mice and control gfp lines. Surprisingly there is a large number of neuronal like activations within the control gfp expressing lines in figure 3b. It is also unclear that these small potential neuronal activations are mapped to known sensory or neural anatomical motifs again bringing up the concern about functional resolution in vivo and spreading of signal which is known to occur (first para concern too). different control lines were employed, it is not clear that the fluorescence properties of all the lines is indeed comparable. If the glio mice have a poorer signal to noise ratio it could potentially recover more artifacts from them than a brightly expressing neuronal line. Ideally the gfp control lines should be better matched to the neuronal signal expected lines (snap25) in terms of laminar expression profile (layer specific) and also absolute intensity and SNR properties.

This could be accomplished using a similar neuronal expressing line for gfp alone or some controls to show that these are somehow comparable.

Figure 4 shows the distribution of recovered activations within the different lines of mice. The figure is notable in that the artifact regions nicely aligned with the expected midline. there are some assumptions about the time course of vascular components across mouse lines. Here the different gfp lines should be potentially measuring the same thing but the time courses can be quite different since they are different cell populations and compartments.

Figure 6 the summing of the different components is intriguing but the authors again need to show that the components exhibit some expected independence by showing time plots from additional adjacent regions. One could also imagine examining some correlational structure for doing some functional mapping such as retinotopic using methods that have previously been employed including the work of Waters (elife) and others to map subdomains.

Figure 7 is welcome in that the maps seem to resemble what is expected for large-scale functional and anatomical relationships in the cortex that have been previously reported. However many of the intra cortical correlations raise some issues such as if the olfactory bulb regional correlation is very high how could one pull out separate areas? There could also be a bit better description for how the small regions are inserted into the macro clusters?

Panel f would be nice to have better color coding of the specific areas and there should be more discussion about the significance of these results. It is not necessarily surprising that an individual animal would be more similar to itself than another mouse because of issues of rising with the alignment or quality of the window preparation. How much of this is animal specific brain activity versus the conditions around the imaging of different mice?

**Have the authors made all data and (if applicable) computational code underlying the findings in their manuscript fully available?**

Reviewer #1: Yes

Reviewer #2: **No: **data will be deposited upon publication

PLOS authors have the option to publish the peer review history of their article (what does this mean?). If published, this will include your full peer review and any attached files.

Reviewer #1: No

Reviewer #2: No
---

## [Decision Letter · Decision Letter 1]

9 Apr 2023

Dear Dr Mullen,

We are pleased to inform you that your manuscript 'Data-driven segmentation of cortical calcium dynamics' has been provisionally accepted for publication in PLOS Computational Biology.

Best regards,

Frédéric E. Theunissen

Academic Editor

PLOS Computational Biology

Lyle Graham

Section Editor

PLOS Computational Biology

Dear Dr. Mullen and co-authors,

Thank you for carefully addressing the reviewer's comments. Reviewer 1 has a few minor suggestions/typos that you should fix in your final submission. It is clear that your contribution will be very useful to the Ca Imaging community.

Best,

Frederic Theunissen

Reviewer's Responses to Questions

**Comments to the Authors:**

Reviewer #1: Weiser and colleagues have successfully addressed all the suggestions previously made. The manuscript makes an original contribution to the scanty literature on the biophysics of mesoscale calcium imaging analysis. In addition, it pushes forward the need for a proper interpretation and understanding of the signal.

In this revised version, the authors expanded the quantitative exploration of the method and it is now more clear and solid. This is a key part of the manuscript since it provides a numerical reference for calcium imaging users about the relationship between the performance of the analysis and the spatial and temporal resolution of the images.

I appreciate that the authors now speculate about the biological interpretation of the independent components. They assume “that the calcium signals collected from structure populations of neurons will produce repetitive and consistent network activation”. Although this is probably a strong assumption, it sets an interesting starting point for exploring their results. Even more important, it can open a scientific debate about the validity of the assumption of statistical independence in the calcium imaging analysis. In any case, the authors are aware and discuss the possible limitations of the method.

The revised validation of the domain maps adds consistency to their previous results and increases the confidence in this kind of map as an alternative when there is no reference map available.

Minor points:

-The authors have removed the terms eigenvector/eigendecomposition from the text when they were not necessary. On this regard, the authors should check supplementary figure 5.

-In the discussion, lines 940 and 941, the use of mean and the median is confusing.

-In the caption of figure 10 section E, the references for top and bottom should be left and right.

**Have the authors made all data and (if applicable) computational code underlying the findings in their manuscript fully available?**

Reviewer #1: Yes

PLOS authors have the option to publish the peer review history of their article (what does this mean?). If published, this will include your full peer review and any attached files.

Reviewer #1: No

---

## [Editor Report · Acceptance letter]

27 Apr 2023

PCOMPBIOL-D-22-01366R1 

Data-driven segmentation of cortical calcium dynamics

Dear Dr Mullen,

I am pleased to inform you that your manuscript has been formally accepted for publication in PLOS Computational Biology. Your manuscript is now with our production department and you will be notified of the publication date in due course.

With kind regards,

Timea Kemeri-Szekernyes
